Corrected: Author correction

# Structural insights into chaperone addiction of toxin-antitoxin systems

Valérie Guillet[1], Patricia Bordes[2], Cécile Bon[1], Julien Marcoux [1], Virginie Gervais[1], Ambre Julie Sala[2], Suzana Dos Reis[1], Nawel Slama[2], Israel Mares-Mejía[1], Anne-Marie Cirinesi[2], Laurent Maveyraud [1], Pierre Genevaux[2] & Lionel Mourey [1]

SecB chaperones assist protein export by binding both unfolded proteins and the SecA motor. Certain SecB homologs can also control toxin-antitoxin (TA) systems known to modulate bacterial growth in response to stress. In such TA-chaperone (TAC) systems, SecB assists the folding and prevents degradation of the antitoxin, thus facilitating toxin inhibition. Chaperone dependency is conferred by a C-terminal extension in the antitoxin known as chaperone addiction (ChAD) sequence, which makes the antitoxin aggregation-prone and prevents toxin inhibition. Using TAC of *Mycobacterium tuberculosis*, we present the structure of a SecB-like chaperone bound to its ChAD peptide. We find differences in the binding interfaces when compared to SecB–SecA or SecB-preprotein complexes, and show that the antitoxin can reach a functional form while bound to the chaperone. This work reveals how chaperones can use discrete surface binding regions to accommodate different clients or partners and thereby expand their substrate repertoire and functions.

[1] Institut de Pharmacologie et de Biologie Structurale, IPBS, Université de Toulouse, CNRS, UPS, 31077 Toulouse, France. [2] Laboratoire de Microbiologie et de Génétique Moléculaires, Centre de Biologie Intégrative (CBI), Université de Toulouse, CNRS, UPS, 31062 Toulouse, France. Correspondence and requests for materials should be addressed to P.G. (email: pierre.genevaux@ibcg.biotoul.fr) or to L.M. (email: lionel.mourey@ipbs.fr)

Universally conserved molecular chaperone machines maintain cellular protein homeostasis by preventing protein misfolding and aggregation, off-pathways in protein biogenesis. In bacteria, the ATP-independent homo-tetrameric chaperone SecB (69 kDa) assists protein export by binding to presecretory proteins and delivering them post-translationally to the Sec translocon at the inner membrane[1]. SecB binds its preprotein clients in a one to one ratio and maintains them in nonnative translocation competent state, protected from premature folding and recognition by cytosolic proteases[2]. Although SecB has no apparent specificity for signal sequences, in vitro peptide scan analysis revealed preferences for stretches of nine amino acids enriched in aromatic and basic residues present in protein clients[3,4]. SecB can accommodate long fragments of up to 150 amino acids in length within polypeptide substrates[5]. The recently determined nuclear magnetic resonance (NMR) structures of SecB bound to its full-length native substrates maltose-binding protein (MBP) and alkaline phosphatase (PhoA) confirmed that SecB possesses a remarkably large substrate-binding surface (about 7600 Å$^2$) where the client protein wraps all around the chaperone in an unfolded conformation, occupying the entire SecB substrate-binding surface[6].

In addition to its preprotein clients, SecB specifically interacts with SecA, the essential ATPase motor subunit of the Sec translocon[7]. Upon interaction, SecB passes on its client to SecA, which subsequently promotes preprotein translocation through the SecYEG pore. SecA-SecB interaction mainly involves the C-terminal zinc-binding region of SecA and the negatively charged surface formed by the β-sheets on both sides of the SecB tetramer[8]. However, additional contacts between the C-terminal helix of SecB and the N-terminal region of SecA have been found[9]. Although the substrate transfer mechanism remains unknown, the surfaces of SecB that interact with the preprotein and with SecA overlap significantly, thus likely facilitating substrate transfer from the chaperone to the translocase.

Besides the generic role of SecB in protein export, SecB homologs are frequently associated (in about 7% of cases) with stress-responsive type II toxin–antitoxin (TA) systems, together forming functional tripartite TA-chaperone (TAC) systems[10]. Classical type II TA systems are small genetic modules encoding two cytosolic proteins: a poisonous toxin and a less stable and protease-sensitive cognate antitoxin, which binds and inhibits the toxin. Under normal growth conditions, the antitoxin-bound toxin is inactive and the bacteria grow normally. It is believed that under certain stress conditions the less stable antitoxin is degraded by proteases and that the resulting free active toxin targets essential cellular processes such as protein synthesis or DNA replication, thus inhibiting growth until normal growth conditions resume[11,12]. TA systems are also widely distributed on mobile genetic elements, including plasmids, where they contribute to their stability. In this case, when the TA-containing plasmid is lost, de novo synthesis of the antitoxin stops and the reservoir of cytosolic antitoxin is rapidly degraded, which results in toxin activation and cell growth inhibition[13]. Control of bacterial growth by toxins has been associated with various cellular processes, including stabilization of genomic regions, protection against foreign DNA, biofilm formation, control of the stress response, bacterial virulence and persistence[14,15]. Although an involvement of TA systems in persistence was found for several bacteria[16–19] their contribution to Escherichia coli K-12 drug persisters has not been demonstrated[20,21]. In tripartite TAC systems, the SecB chaperone encoded within the same operon as the TA genes, directly interacts with the highly unstable TAC antitoxin and prevents its aggregation and degradation, thus facilitating subsequent inhibition of the toxin[22].

The human pathogen Mycobacterium tuberculosis possesses such a TA-associated SecB chaperone, named Mtb-SecB$^{TA}$, which specifically controls the Mtb-HigB1 (toxin)-HigA1 (antitoxin) TA pair[12,22]. Mtb-HigB1, which likely belongs to the RelE toxin superfamily of ribonucleases, was shown to severely inhibit E. coli, M. smegmatis, M. marinum, and M. tuberculosis growth[12]. Noticeably, Mtb-TAC genes were shown to be significantly induced in response to several stresses relevant for M. tuberculosis pathogenesis. This includes DNA damage[23], heat shock[24], nutrient starvation[25], hypoxia[26], host phagocytes[27] and antibiotic-induced persistence[28], suggesting that the TAC system could contribute to the stress adaptive response and/or to the virulence of this bacterium. Yet, such a role for TAC remains to be demonstrated.

We have recently shown that in contrast with chaperone-independent TA systems, the Mtb-HigA1 antitoxin of TAC possesses a C-terminal extension, named ChAD (chaperone addiction), which makes the antitoxin more aggregation-prone and renders it chaperone-dependent. In this case, the ChAD extension is specifically recognized by the Mtb-SecB$^{TA}$ chaperone, which protects it from both aggregation and degradation[29].

In this work, we provide biochemical and structural insights into the mechanism of chaperone addiction. Using TAC of M. tuberculosis as model system, we solved the crystal structure, at high resolution, of a TA-associated SecB chaperone bound to the ChAD peptide of its cognate antitoxin. The structure reveals major differences in binding mode and substrate occupancy when compared to SecB–SecA or SecB–substrate complexes. Further extensive mutagenesis of contact surface residues revealed several previously unidentified residues essential for TA-control by Mtb-SecB$^{TA}$ chaperone in vivo. Finally, we showed that the antitoxin can reach a functional form while bound to the chaperone and that ChAD peptide may efficiently destabilize the interface between the antitoxin and the chaperone in vitro and specifically activate the Mtb-HigB1 toxin in vivo.

## Results

**Three-dimensional structure of Mtb-SecB$^{TA}$ in complex with ChAD.** In contrast with the classical export chaperone SecB, which binds and wraps unfolded presecretory protein substrates and maintains them in an unfolded conformation compatible with translocation through membrane[6], previous work revealed that the Mtb-SecB$^{TA}$ chaperone of TAC is capable of binding to its aggregation-prone Mtb-HigA1 antitoxin substrate to facilitate, and thus not prevent, its folding in the cytosolic space. This strongly suggests a different binding mode of SecB-like chaperones with their antitoxin clients. To investigate the SecB substrate-binding property, we have solved the structure of Mtb-SecB$^{TA}$ together with the short C-terminal fragment of the Mtb-HigA1 antitoxin (named ChAD), which was found as the main site of interaction with the chaperone, both in vivo and in vitro[29,30]. The structure comprises a Mtb-SecB$^{TA}$ homo-tetramer (the subunits, termed A–D, contain 136–138 residues of the 181 amino acids found in the Mtb-SecB$^{TA}$ sequence) to which three ChAD peptides (corresponding to sequence $_{104}$EVPTWHRLSSYRG$_{116}$ of Mtb-HigA1 named C4 in a previous study[29]; subunits E–G), 2 dimethyl sulfoxide (DMSO) molecules, and 3 Ca$^{2+}$ ions are bound (Table 1 and Fig. 1a). The Mtb-SecB$^{TA}$ structure resembles that of SecB for which four structures have been solved: the X-ray structures of Haemophilus influenzae SecB alone[31] and in complex with the C-terminal 27 residues of SecA[8], the X-ray structure of E. coli SecB alone[32], and the NMR structures of its complexes with alkaline phosphatase and maltose-binding protein in their unfolded states[6] (Supplementary Fig. 1a). The three-dimensional fold of Mtb-SecB$^{TA}$ consists of a

## Table 1 Crystallographic data collection and refinement statistics

| | *Mtb*-SecB[TA]/HigA1(104–116) PDB code 5MTW |
|---|---|
| *Data collection* | |
| Diffraction source | ESRF ID29 |
| Space group | $P2_12_12_1$ |
| Unit cell $a$, $b$, $c$ (Å) | 86.86, 89.96, 91.34 |
| Unit cell $\alpha$, $\beta$, $\gamma$ (°) | 90.00, 90.00, 90.00 |
| Resolution range (Å) | 64.09–1.835 (1.867–1.835)[a] |
| No. of unique reflections | 63,188 (3135) |
| Completeness (%) | 99.9 (99.6) |
| Redundancy | 6.5 (6.7) |
| $\langle I/\sigma I\rangle$ | 13.7 (2.3) |
| $R_{merge}$ | 0.093 (0.796) |
| $CC_{1/2}$ | 0.999 (0.338) |
| *Refinement* | |
| Resolution range (Å) | 64.09–1.835 (1.883–1.835)[a] |
| No. of reflections (work/test) | 59,790/3397 (4325/275) |
| No. of molecules/AU[b] | 1 *Mtb*-SecB[TA] tetramer + 3 bound *Mtb*-HigA1(104–116) peptides |
| $R_{work}/R_{free}$ | 0.212/0.265 (0.330/0.354) |
| Chain/no. of residues/missing residues | A/138/1–8, 43–44, 82–96, 164–181 |
| | B/138/1–5, 43–46, 81–96, 164–181 |
| | C/138/1–12, 82–94, 114–116, 167–181 |
| | D/136/1–9, 81–95, 114–115, 163–181 |
| | E/12/116 |
| | F/12/116 |
| | G/12/116 |
| *No. of nonhydrogen atoms* | |
| All atoms | 4660 |
| Protein atoms | 4282 |
| Peptide atoms | 318 |
| Other ligands | 8 |
| Ions | 3 |
| Water molecules | 49 |
| *Average B-factors (Å²)* | |
| All atoms | 34.7 |
| Protein atoms | 34.3 |
| Peptide atoms | 39.7 |
| Other ligands | 63.5 |
| Ions | 45.7 |
| Water molecules | 32.6 |
| *RMS deviations* | |
| Bond lengths (Å) | 0.019 |
| Bond angles (°) | 1.941 |
| *Ramachandran plot (%)* | |
| Most favored regions | 93.9 |
| Allowed/disallowed | 6.1/0.0 |

[a]Values in brackets are for the highest-resolution shell.
[b]Asymmetric unit.

six-residue long N-terminal helix (αN) followed by a four-stranded antiparallel β-sheet (β1–β4), each strand contributing 13–16 residues, and a 30-residue long C-terminal helix (α1) that is prolonged by an extended peptide segment running parallel to α1 (Fig. 1b and Supplementary Fig. 1b). Missing residues are found in the majority at the N and C termini (5–12 and 15–19 residues missing, respectively) and in the loop between β3 and β4 (13–16 residues missing), in all subunits. The superposition of the subunits making up the *Mtb*-SecB[TA] tetramer led to a root mean square deviation (rmsd) of Cα atoms ranging between 0.4 (subunit C vs. B) and 1.2 Å (subunit C vs. A). This is in line with the NMR characterization of SecB, which revealed two pairs of spectroscopically equivalent subunits formed on the one hand by subunits A and D and on the other hand by subunits B and C[6]. Largest deviations correspond to spatial locations lining regions

with missing residues, which in turn correlates with high *B*-factors, thus indicating structural flexibility.

The *Mtb*-SecB[TA] tetramer consists of a dimer of dimers as already described for SecB[31,32]. Two subunits associate through their β1 strands to form β-dimers (A–B and C–D), resulting in an eight-stranded antiparallel β-sheet (Fig. 1a). In the tetramer, the A–B and C–D β-dimers are stacked up through their helices α1 at an angle of ca. 45° (Fig. 1a). The dimer and dimer–dimer interface areas are in the same range: 739/728 Å² for A–B/C–D vs. 820/704 Å² for A–C/B–D (Supplementary Table 1). It is noteworthy that the tetramer displays internal dihedral ($D_2$) symmetry that includes three perpendicular twofold symmetry axes. However, not all among the three possible superimpositions through rotations around these axes are equivalent. Only one permutation (DCBA) gives a rmsd value of 0.6 Å after superposing 522 Cα atoms to the reference tetramer (ABCD), whereas the two others (BADC and CDAB) led to much higher rmsd values of 3.5 Å. The DCBA permutation is generated through rotation around the twofold axis that lies between the dimer–dimer interface and runs nearly perpendicular to helices α1 (Fig. 1a). This distorted dihedral symmetry is also found in the previously determined SecB structures (PDB entries 1FX3, 1OZB, 1QYN, and 5JTL), but has never been described before. Molecular dynamics calculations performed on the structure suggest that this distorted arrangement is stable over the 60-ns simulation time, and that once trapped into such a conformation, no further fluctuation does occur (Supplementary Fig. 2a).

It has recently been reported that diffraction-quality crystals of *Mtb*-SecB[TA] could only be grown at high concentrations of DMSO. Indeed, the use of DMSO improved initial needle-like crystals but was not sufficient to resolve the structure[33]. Adding 5–10% DMSO in the protein and/or crystallization buffer played a role with respect to nucleation but did not markedly influence the diffraction limit of *Mtb*-SecB[TA]/ChAD crystals. We also noted that the presence of calcium ions was important for the crystallization of the complex. Three $Ca^{2+}$ ions were localized based on electronic density and geometric criteria, and calcium binding occurs in subunits A, B, and D at the same position (Fig. 1a). Indeed, all three ions interact with main-chain nitrogen atom and side-chain oxygen atom of S65, with the main-chain oxygen atom of Y111, and with the carboxylate of D110 thereby linking the N- and C-termini of strands β3 and β4, respectively. The lack of calcium binding to subunit C is due to crystal packing constraint where loop β4-α1 of a symmetry mate partly occupies the site.

**Comparison of *Mtb*-SecB[TA] and SecB structures.** Pairwise comparison of the *Mtb*-SecB[TA] crystallographic tertiary structure with those of *Hi*-SecB and *Ec*-SecB gave rmsd values around 2.0 Å for about 110 Cα atoms of residues sharing 15% identity. In comparison, superimposition of the *Hi*- and *Ec*-SecB subunits led to rmsd values less than 1.0 Å for ca. 130 residues sharing 59% identity. There are three major differences between the *Mtb*-SecB[TA] tertiary structure and its SecB counterparts (Fig. 1b and Supplementary Fig. 1b): (i) *Mtb*-SecB[TA] has a longer N terminus that adopts a helical fold (αN), (ii) it has a 12- to 14-residue long insertion in the loop between β3 and β4, which is disordered in the structure, and (iii) its C terminus is shorter and could not be traced beyond residue number 164. In contrast, the C-terminal ends of *Hi*- and *Ec*-SecB are better defined in density where they form helix α2. We verified by mass spectrometry (MS) of dissolved crystals that no proteolytic cleavage, which could account for the missing C-terminal residues, had occurred in the crystalline state. Structural dynamics might then be responsible for the absence of electron density for the C-terminal residues of

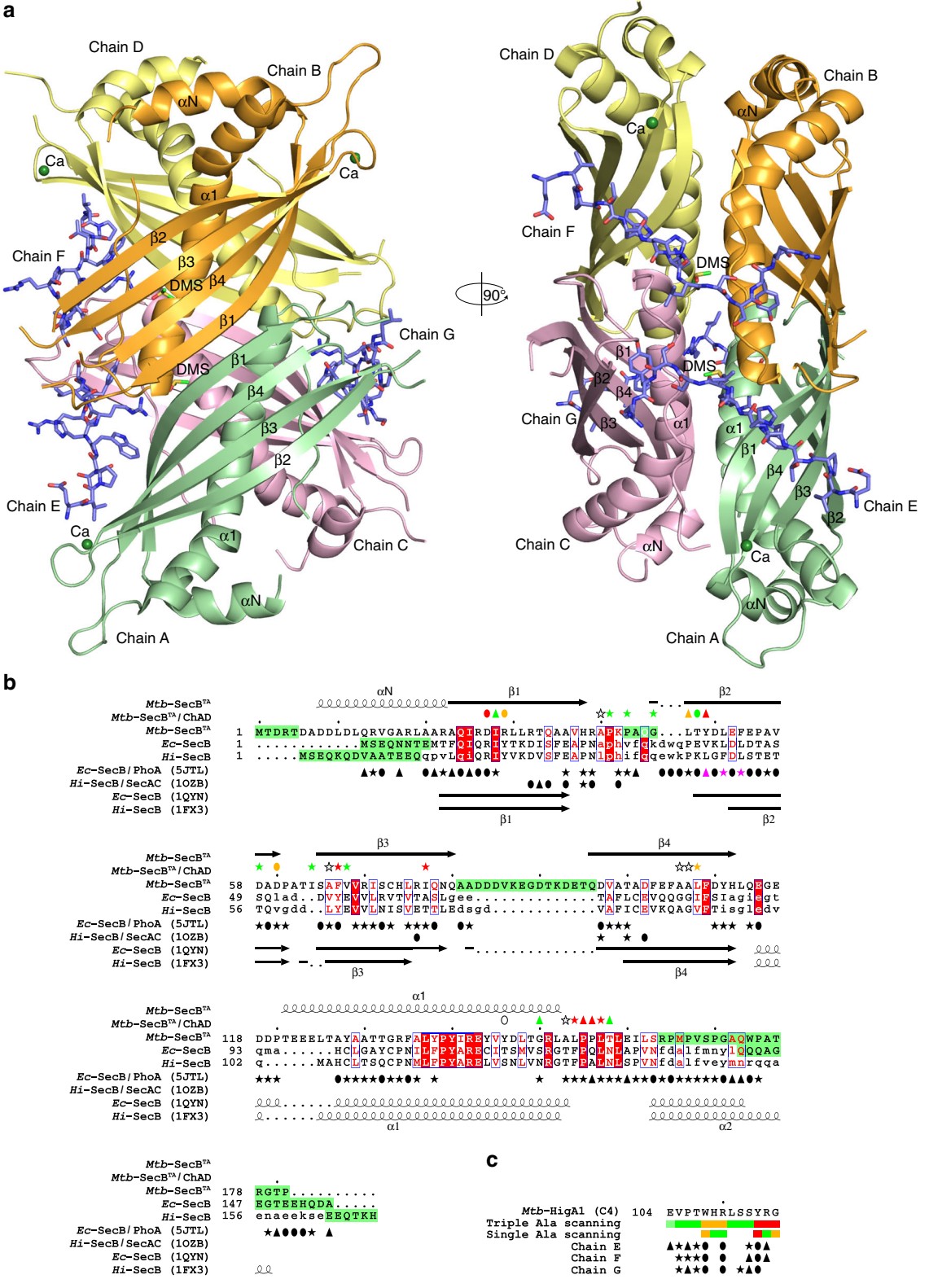

SecB<sup>TA</sup> and the location of a C-terminal α2 helix as found in *Hi*- and *Ec*-SecB. Such flexibility might be fostered by the proximal location of the N-terminal helix, whose position is dictated for some of the subunits by crystal packing constraints, and which otherwise would induce steric clashes (Supplementary Fig. 1b). Other differences between the *Mtb*-SecB<sup>TA</sup> and SecB structures occur in loops β1–β2, β2–β3, and β4–α1 (Fig. 1b and Supplementary Fig. 1). Superimposing the crystal structure of *Mtb*-

SecB<sup>TA</sup> tetramer with those of *Ec*- and *Hi*-SecB gave rmsd values around 2.0 Å. In comparison, superposing the *Ec*- and *Hi*-SecB tetramers led to rmsd values of 0.9 Å. Although the tetramers are highly similar in structure, important variations were found when comparing the dimer and dimer-dimer interface areas (Supplementary Table 1). Whether these structural differences are important for TA control, export function or protein stability is unclear. The fact that all the TAC chaperones tested so far could

**Fig. 1** Overall structure of the *Mtb*-SecB$^{TA}$/ChAD complex and determinants of SecB ligand specificity. **a** The A–D subunits forming the *Mtb*-SecB$^{TA}$ tetramer are respectively colored in green, orange, violet, and yellow whereas ChAD peptides (chains E–G) are in slate blue. Calcium ions are shown as green spheres. DMSO molecules (labeled DMS) are shown as a stick representation. Left and right views are separated by a 90° rotation around a vertical axis. Secondary structures have been labeled for the A–B (left) and A–C (right) dimers. **b** Structure-based sequence alignment. Sequence similarity is indicated by red letters, whereas sequence identity is indicated by white letters on a red background. Aligned and unaligned residues are displayed in uppercase and lowercase, respectively, taking *Mtb*-SecB$^{TA}$ as reference. Secondary structure elements as deduced from the crystal structures of *Mtb*-SecB$^{TA}$/ChAD, *Ec*-SecB (PDB code 1QYN), and *Hi*-SecB (1FX3) are indicated. Residues absent from the structures are on a green background. Residues involved in protein–protein interface of the *Mtb*-SecB$^{TA}$/ChAD, *Ec*-SecB/PhoA (5JTL), and *Hi*-SecB/SecAc (1OZB) complexes are indicated by symbols: solid ellipses and triangles, residues involved in polar interactions through side-chain and main-chain atoms, respectively; stars, residues having ≥50% of their surface area buried upon van der Waals interactions. For *Mtb*-SecB$^{TA}$, color coding of the symbols is according to the impact of alanine scanning on the ability to maintain TA control from green (no effect) to orange (medium effect) to red (most severe effect). Residues that were not tested (i.e., Ala residues and Y146, which was too strongly expressed) are designated by unfilled symbols. The V40A/L42A/L44A triple substitution that causes a 40-fold reduction in the affinity of *Ec*-SecB for PhoA is in purple. **c** Sequence of the *Mtb*-HigA1 ChAD peptide. The rectangles below the sequence represent the amino acids that have been changed for alanine as previously reported[29]. Green rectangles represent mutations without apparent phenotype with respect to their chaperone dependence in vivo; orange, mutations with partial inactivation; red rectangles, mutations with the most severe phenotype. Interfacing residues with *Mtb*-SecB$^{TA}$ are depicted using the same symbol scheme as in **a**

efficiently replace *Ec*-SecB export function, regardless of the sequence similarities[10,29], and that single-point mutations in substrate-binding area were sufficient to redirect *Ec*-SecB toward *Mtb*-HigBA1 without affecting its function in protein export[30], suggest that such differences may not be critical for TA control or export.

*Mtb*-SecB$^{TA}$ and related *Ec*- and *Hi*-SecB are acidic proteins that remarkably share a strictly identical theoretical pI value of 4.3. In accordance, analysis of the topography and charge distribution revealed that they display very electronegative molecular surfaces but differences between the SecB proteins and *Mtb*-SecB$^{TA}$ are noticeable (Supplementary Fig. 3). Major differences pertain in fact to negative charge distribution. For instance, there is a strong electronegative cluster right in the middle of the SecB eight-stranded β-sheet, which arises from side-chain clustering of residues D20/D27, E24/E31, and E77/E86 (*Ec*-SecB/*Hi*-SecB numbering), which is important for interaction with SecA C-terminal region. In *Mtb*-SecB$^{TA}$, these residues are replaced by R32, A36, and D102, respectively. The fact that *Mtb*-SecB$^{TA}$ is less efficient than *Ec*-SecB when tested for SecA-dependent function in *E. coli* suggests that such modified electrostatic interface might be responsible for the reduced complementation in vivo.

Another important difference is found in a groove delineated by the N-terminus of strand β1, the β1–β2 loop, helix α1, and the extended C-terminal segment. This groove is neutral or slightly positively charged in SecB whereas it is electronegative in *Mtb*-SecB$^{TA}$, involving the D27, E143, and D147 residues and the positively charged R151. In contrast, only one acidic residue is found in SecB, E112/E121, which corresponds to E143 of *Mtb*-SecB$^{TA}$, and is counterbalanced by R120/R129 equivalent to R151 in *Mtb*-SecB$^{TA}$.

**Mtb-SecB$^{TA}$/ChAD interaction.** In order to investigate how the chaperone specifically recognizes the C-terminal chaperone-addiction region of the *Mtb*-HigA1 antitoxin, key residues involved in the interaction between the ChAD peptide (*Mtb*-HigA1 $_{104}$EVPTWHRLSSYRG$_{116}$) and the *Mtb*-SecB$^{TA}$ chaperone substrate-binding surface were analyzed in detail. Note that this small peptide represents the main region of the ChAD extension of *Mtb*-HigA1 that is recognized by the chaperone, both in vivo and in vitro[29]. In the structure, there are three bound ChAD peptides (chains E–G) per tetramer (Fig. 2a). For all three peptides, there was no adequate electron density to build the last residue (G116) and the side chain of R115 is also missing in peptide G (Supplementary Fig. 4). Binding occurs on the thinner side of the *Mtb*-SecB$^{TA}$ chaperone tetramer. On one side, two

peptides (chains E and F) are bound in a symmetrical way (related by a twofold axis) and adopt strictly the same extended conformation with a rmsd of 0.6 Å based on main-chain atoms. The third peptide (chain G) binds on the opposite side and also adopts an extended conformation identical from residues 104–108 (rmsd of 0.5 Å on backbone atoms) but differs in main-chain ($\varphi,\psi$) angles from residues 109–114 (rmsd of 2.1 Å). It is noteworthy that peptide G differs in conformation from peptides E and F to conserve the same type of interactions with the protein, and molecular dynamics (MD) calculations indicate that such binding mode is preserved during the 60 ns of simulation (Supplementary Fig. 2b). On the other hand, no sufficient interpretable electron density could allow the building of a symmetrical equivalent of peptide G (Supplementary Fig. 4), which because of steric clashes due to the distorted D$_2$ symmetry would not be possible without structural rearrangement. Each bound peptide makes contact with three protein subunits in a similar way, with some variations (Fig. 1c). A residue-per-residue detailed description of the interactions is given in Supplementary Table 2.

In agreement with previous in vivo data[29,30], the structure reveals that three ChAD residues play a major structural role in complex formation, namely W108, R110, and Y114. *Mtb*-HigA1 (W108) makes strong interactions with the protein (Fig. 2b). It is involved in hydrogen bonding with D27 located on strand β1. In addition, its indol ring is completely buried in a protein cavity delineated by residues D27, R29, V68, and L108, from one *Mtb*-SecB$^{TA}$ subunit, and A153 (at the C terminus of α1) from another *Mtb*-SecB$^{TA}$ subunit. It is also involved in cation-π interactions with R29 side chain. *Mtb*-HigA1(R110) makes strong polar interactions with D27 and I28 from one subunit, and with G150 (at the C terminus of α1) from another subunit (Fig. 2b). van der Waals interactions with a third protein subunit also contributes to the steady anchoring of *Mtb*-HigA1(R110). *Mtb*-HigA1(Y114) is hydrogen bonded to the protein through interaction with P155. In addition, its phenol group perfectly fits a gorge delineated by residues L47, Y49, L154, P155, P156, and L157 (Fig. 2c).

**Mutagenesis of Mtb-SecB$^{TA}$ residues involved in ChAD binding.** In order to investigate whether the interaction between *Mtb*-SecB$^{TA}$ and the antitoxin was relevant in vivo, all residues of *Mtb*-SecB$^{TA}$ involved in polar interactions with the ChAD peptide, either through side-chain or main-chain atoms, and those involved in nonbonding interactions through burying more than 50% of their solvent accessible surface (Fig. 1b and Supplementary Table 2) were systematically mutated to alanine residues. The resulting variants were then tested for both their

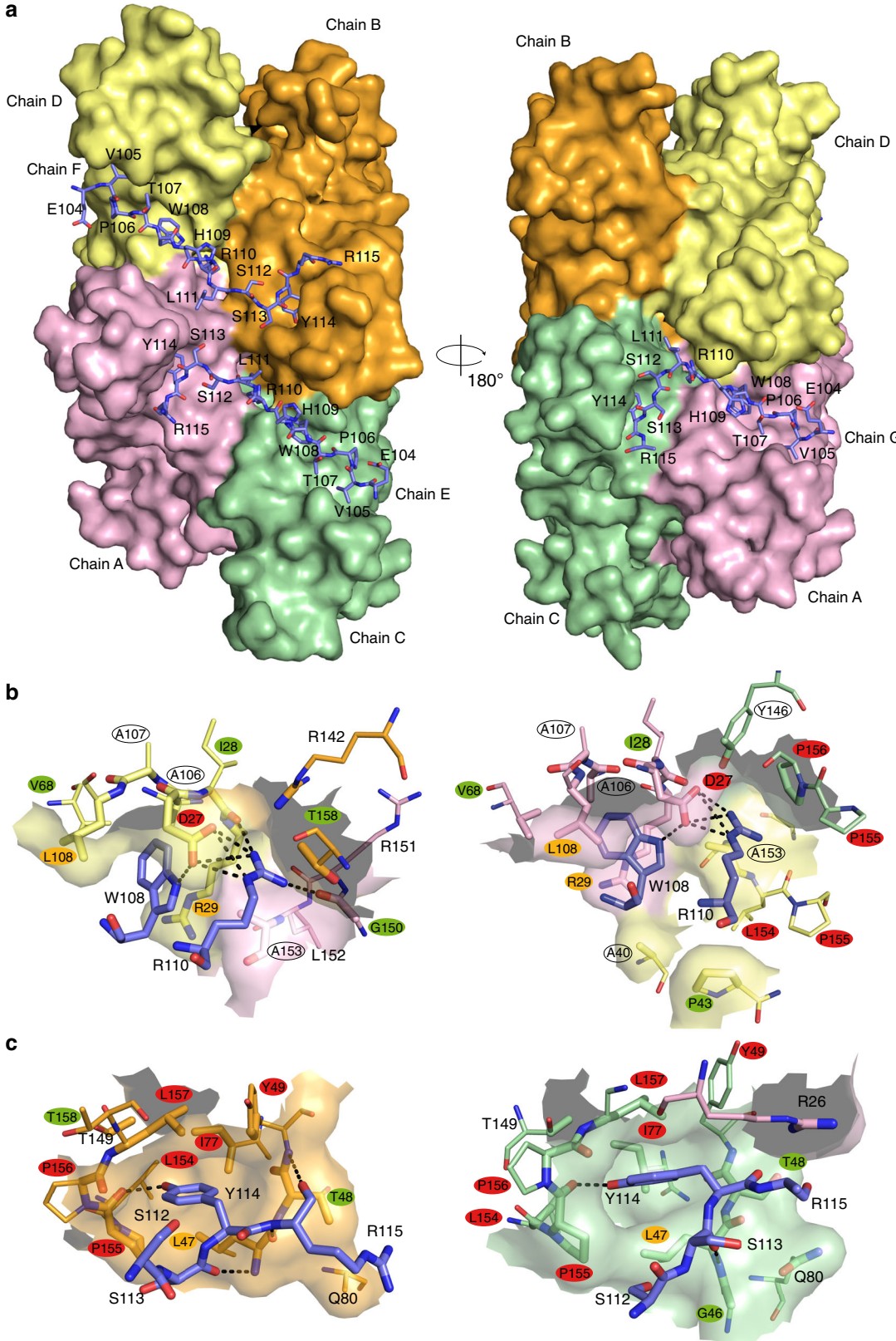

**Fig. 2** ChAD binding on *Mtb*-SecB^TA. **a** Molecular surface of the tetramer. The three bound peptides are shown as a stick representation with carbon, nitrogen, and oxygen atoms in slate blue, blue, and red, respectively. The protein and peptide chains as well as ChAD residues have been labeled. Left and right views are separated by a 180° rotation around a vertical axis. **b** Close-up view centered on ChAD residues W108 and R110. **c** Close-up view centered on ChAD residue Y114. In **b** and **c**, *Mtb*-SecB^TA atoms found within 5 Å of the peptides are shown as enlarged sticks and depict the ChAD-binding site shown as a semitransparent surface. Polar interactions are represented by black dotted lines. Views on the left (resp. right) are for subunit F (resp. G). Color coding of interacting residues is the same as in Fig. 1b

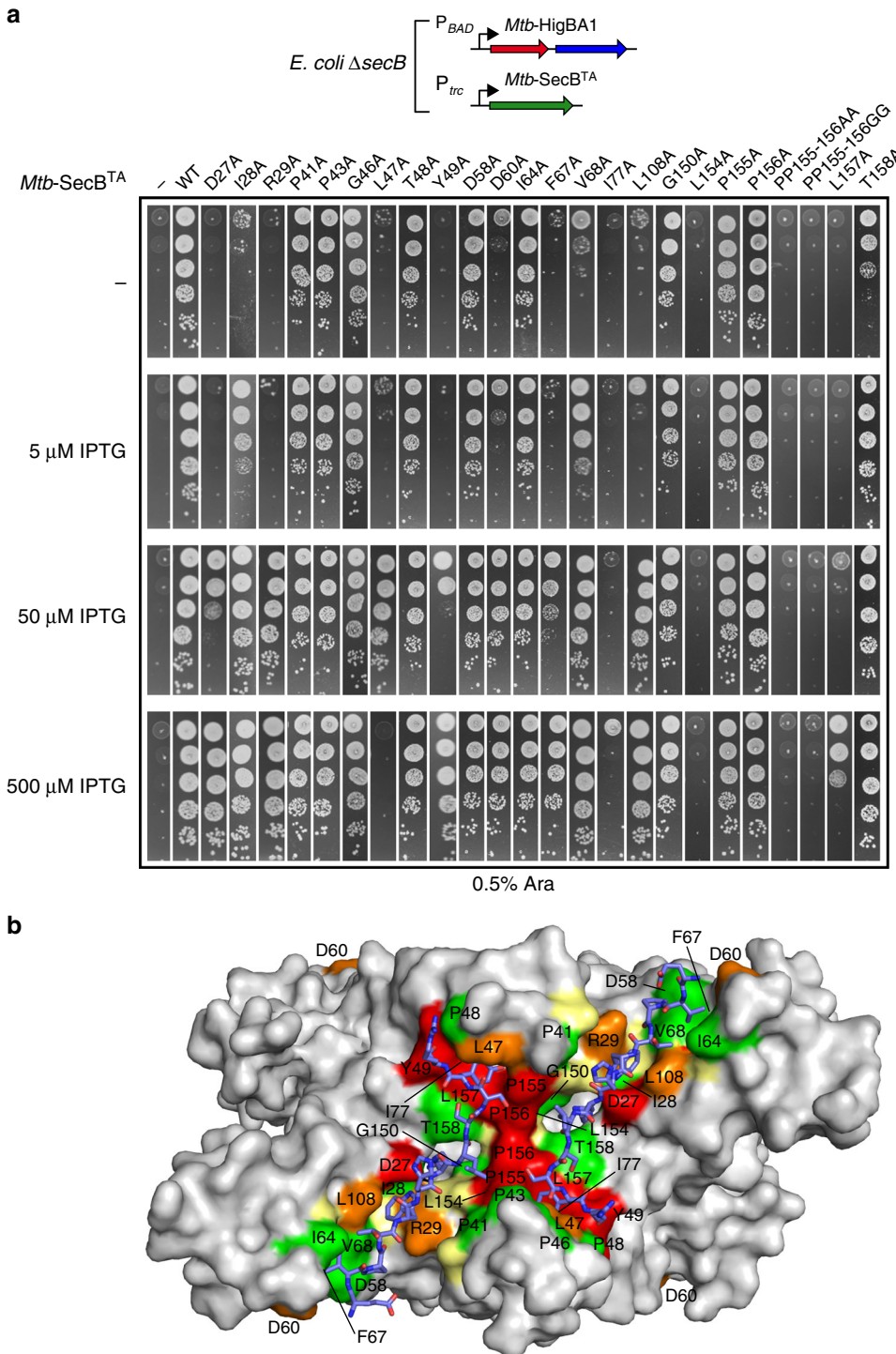

**Fig. 3** In vivo functions of TA-directed *Mtb*-SecB^TA mutants. **a** Suppression of *Mtb*-HigBA1 toxicity by *Mtb*-SecB^TA mutants. *E. coli* W3110 Δ*secB* transformants containing the p29SEN-based *Mtb*-SecB^TA chaperone mutants and pK6-*Mtb*-HigBA1 were grown to mid-log phase, serially diluted and spotted on LB ampicillin kanamycin agar plates with or without IPTG (to express *Mtb*-SecB^TA) and arabinose (to express *Mtb*-HigBA1) as indicated. Plates were incubated overnight at 37 °C. Spot tests have been performed in triplicates. **b** Localization of the mutations on the *Mtb*-SecB^TA/ChAD structure. The molecular surface of the tetramer is represented in light gray. Color coding of interacting residues is the same as in Fig. 1b. Residues that were not tested (i.e., Ala residues and Y146) are colored in pale yellow

ability to control the toxin (Fig. 3) and to functionally replace the canonical *E. coli* export chaperone SecB with respect to its generic chaperone function at low temperature and in the presence of novobiocin antibiotic[34,35] (Supplementary Fig. 5). Steady state expression of the *Mtb*-SecB^TA derivatives revealed that all mutants are well expressed when compared to the wild type

(Supplementary Fig. 6). *Mtb*-SecB^TA mutations that most significantly affect TA control were found at positions making strong interactions with ChAD residues shown to be involved in chaperone binding[29], namely residues D27, R29, and L108 interacting with *Mtb*-HigA1(W108), and residues Y49, I77, L154, P155, P156, L157 interacting with *Mtb*-HigA1(Y114) (Fig. 2b, c).

It is noteworthy that *Mtb*-HigA1(G116), whose mutation to alanine was also associated to a defect in vivo[29], could not be resolved in the electron density. Moreover, residues D27 and L154 of *Mtb*-SecB$^{TA}$ are also involved in interactions with R110 of *Mtb*-HigA1 (Fig. 2b). Two other *Mtb*-SecB$^{TA}$ residues, L47 and F67, appeared to be important in vivo. Although not directly interacting with *Mtb*-HigA1(Y114), L47 contributes to the pocket that accommodates the tyrosine phenol ring (Fig. 2c). The role played by F67 seems to be more structural since the phenylala-nine ring is oriented toward the interior of the protein thereby contributing to a network of hydrophobic/aromatic interactions. The mutation for an alanine might locally destabilize the protein hydrophobic core and in turn the positioning of strand β3 backbone, which is involved in ChAD interaction. Noticeably, although spread all over the sequence, essential residues of *Mtb*-SecB$^{TA}$ involved in the interaction with ChAD define two major binding spots when mapped on the 3D structure (Fig. 1). The first spot is located on the β-sheet of one of the four tetramer subunits, and involves strands β1–β3. The second spot, which comprises the residues most severely affected by mutations, is found on the protomer of the adjacent β-dimer. It involves the N-terminal tip of strand β2 and residues that follow helix α1 and form the helix connecting loop. Such crosslinking of ChAD peptides at the surface of the *Mtb*-SecB$^{TA}$ tetramer might tighten, or even lock, the dimer–dimer interface and explain the improvement found in both diffraction quality and resolution of the crystals of the *Mtb*-SecB$^{TA}$/ChAD complex vs. those of *Mtb*-SecB$^{TA}$.

All the *Mtb*-SecB$^{TA}$ mutations that most significantly impaired TA control additionally affected the ability to replace the *E. coli* export chaperone SecB in vivo at low temperature of growth, with the exception of Y49A (Supplementary Fig. 5). Remarkably, Y49A significantly improved *Mtb*-SecB$^{TA}$ ability to replace *Ec*-SecB in vivo. Two other mutations, G46A and D58A, slightly improved *Mtb*-SecB$^{TA}$ chaperone generic function but, in contrast with Y49A, without affecting TA control. These three residues are spread along strand β2, G46, and Y49 close to each other at the N-terminal side and D58 at the C terminus. It is noteworthy that *Mtb*-SecB$^{TA}$(Y49) corresponds to *Ec*-SecB(V40) whose mutation as part of the triple amino-acid substitution V40A/L42A/L44A (with all three residues on β2) was shown to cause a 40-fold reduction in the affinity of *Ec*-SecB for PhoA[6]. Whether these mutations affect SecB substrate-binding specificity or its interaction with its SecA partner is not known.

## Specific affinity of ChAD for *Mtb*-SecB$^{TA}$ chaperone.

Several approaches were used to characterize complex formation between *Mtb*-SecB$^{TA}$ and the ChAD peptide in solution and to further confirm structure-function relationships deduced from the X-ray structure and mutagenesis experiments. Isothermal titration calorimetry (ITC), performed under both direct and reverse conditions, and microscale thermophoresis (MST) showed that the chaperone interacts with the ChAD peptide with moderate affinity, with a $K_d$ dissociation constant of $1.9 \pm 0.2$ and $2.6 \pm 0.6$ μM, respectively (Fig. 4a, b). The ITC-derived thermodynamic parameters indicated an enthalpy-driven mode ($-50.1$ kJ mol$^{-1}$) accompanied with an entropic penalty ($-T\Delta S = 18.7$ kJ mol$^{-1}$), likely suggesting formation of hydrogen bonds associated with binding, which is in line with the crystal structure. On the other hand, the stoichiometry obtained by direct and reverse ITC for this specific association is 0.47 and 2.2, respectively, consistent with a model preferentially involving two peptides per *Mtb*-SecB$^{TA}$ tetramer. Extensive small-angle X-ray scattering (SAXS) analysis performed on different protein preparations system-atically revealed that the scattering curves corresponding to *Mtb*-SecB$^{TA}$ and *Mtb*-SecB$^{TA}$/ChAD peptide displayed small but

significant differences (Fig. 4c). Both Guinier and the concentration-independent method[36] were applied to SAXS data and gave respective molecular weight of $84.5 \pm 0.1$, $81.7 \pm 0.4$ kDa for *Mtb*-SecB$^{TA}$ and $89.7 \pm 0.1$, $83.6 \pm 0.3$ kDa for *Mtb*-SecB$^{TA}$/ChAD. The chaperone alone and its complex with ChAD form well folded molecular species, as can be observed from normalized Kratky plots (Fig. 4c, inset), and Guinier analysis gave $R_g$ values of $3.17 \pm 0.02$ and $3.02 \pm 0.02$ nm and maximum dimensions of the particles (Dmax) of $9.5 \pm 0.2$ and $8.8 \pm 0.2$ nm, respectively. This indicates that, at the resolution probed, *Mtb*-SecB$^{TA}$ does not undergo a major conformational change upon ChAD peptide binding but rather displays a slight compaction and a less dynamic behavior. These properties might be related to the better dif-fracting quality of the crystals obtained in the presence of the peptide. In addition, the scattering curve calculated from the *Mtb*-SecB$^{TA}$/ChAD peptide crystal structure, where the missing loops representing 24% of the unbuilt residues have been modeled, led to a rather good, albeit not perfect, agreement ($\chi = 2.6$) with experimental data (Fig. 4c). To further investigate the dynamic behavior of the chaperone, we performed Hydrogen–Deuterium eXchange MS (HDX-MS) on *Mtb*-SecB$^{TA}$ alone and in the pre-sence of ChAD peptide. The kinetics of exchange for *Mtb*-SecB$^{TA}$ alone revealed some highly dynamic and/or accessible regions (namely residues 13–18, 37–51, 59–67, and 161–181, in line with the absence of electron density for the C-terminal residues of *Mtb*-SecB$^{TA}$) compared to more protected regions (residues 68–75, 98–108, and 124–152). Interestingly, a large decrease in deutera-tion upon binding of the ChAD peptide was observed for residues 21–27 and 153–159, which are precisely located along the peptide-binding pocket, and juxtaposing residues 52–57 (Supplementary Fig. 7a). These results also confirm that the chaperone is rather stable upon complex formation with ChAD.

The deconvoluted MS spectrum obtained for *Mtb*-SecB$^{TA}$ under nondenaturing conditions clearly showed the presence of a tetramer at 81,575 Da and up to 7 additional species, separated by 1587 Da, progressively appearing upon titration with the ChAD peptide (Fig. 4d and Supplementary Fig. 8a). Binding of the first four peptides to the tetrameric chaperone corresponds to relatively high-affinity events, as they occur at small peptide:tetramer ratios (<4), whereas binding of the three additional peptides only occurred at high molar ratios (>4), probably due to nonspecific interactions (Supplementary Fig. 8a). Plotting the relative abundance obtained for each species at increasing molar ratios and fitting the corresponding curves (Supplementary Fig. 8b) confirmed this observation, with $K_d$ values in the range 6–17 μM. Incubation of the chaperone with 8 molar equivalents of 13-mer peptides, which also belong to the ChAD sequence but do not contain the main chaperone-binding determinants[29], only showed limited binding to the tetramer, confirming the higher binding specificity observed for the ChAD peptide (Fig. 4e). This low-abundance interaction may be related to some non-specific binding also observed at high molar ratios of ChAD peptide. The nonspecific adduction is a common feature of native MS when working with substrate/ligand concentrations greater than 50 μM[37].

## Full-length *Mtb*-SecB$^{TA}$/HigA1 chaperone/antitoxin complex.

Multiple attempts to crystallize the full-length *Mtb*-SecB$^{TA}$/HigA1 complex were unsuccessful, although such complex could be easily purified in a stable soluble form. Size exclusion chro-matography and multi-angle light scattering (SEC-MALS) clearly indicated that *Mtb*-SecB$^{TA}$ alone primarily forms tetramers while experiments performed on *Mtb*-SecB$^{TA}$ co-expressed with *Mtb*-HigA1 revealed a single-molecular species representing a 4:2 chaperone/antitoxin stoichiometry (Fig. 5a). These results were further confirmed by native MS (114.9 kDa, see next paragraph).

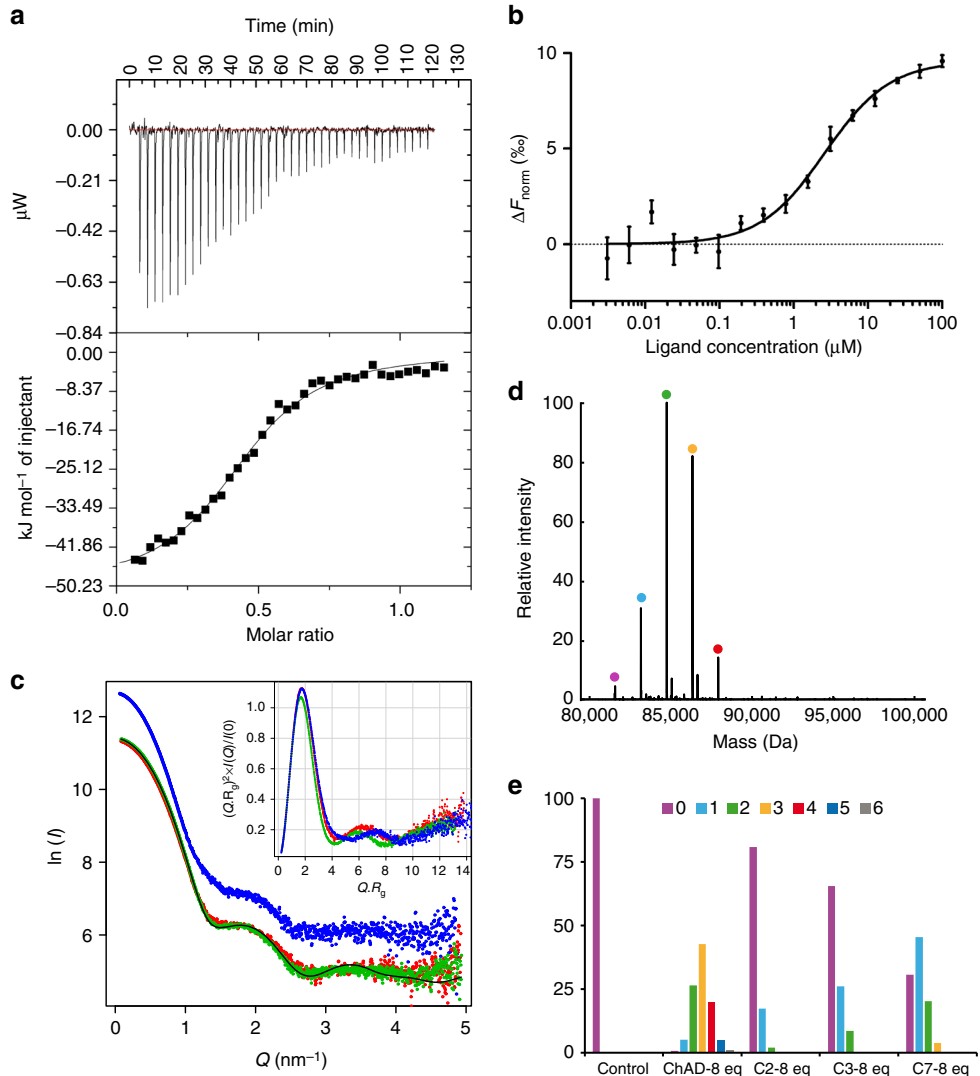

**Fig. 4** Molecular analysis of ChAD binding to *Mtb*-SecB[TA]. **a** ITC profile of ChAD titration into *Mtb*-SecB[TA] (upper and lower parts represent raw and integrated binding heats, respectively). **b** Differences in normalized fluorescence generated by various concentrations of ChAD added to *Mtb*-SecB[TA] during MST analysis and calculated fit. Error bars are as calculated with the NanoTemper analysis software based on independent experimental data. **c** Experimental SAXS data for *Mtb*-SecB[TA] in the unbound state (red line), in the presence of ChAD (green line), and for the complex formed with *Mtb*-HigA1 (blue line) and comparison with the theoretical scattering patterns computed from the *Mtb*-SecB[TA]/ChAD structure (black line). The inset shows the SAXS-derived Kratky plots. **d** Deconvoluted native mass spectrum of *Mtb*-SecB[TA] incubated with 4 molar equivalents of ChAD peptide. The species detected correspond to the *Mtb*-SecB[TA] tetramer (MW 81,575 Da, purple circle) and to the binding of up to four ChAD peptides with MWs of 83,157; 84,746; 86,334 and 87,922 (cyan, green, orange, and red, respectively). **e** Native MS-derived relative abundance of the different species obtained upon titration with 8 molar equivalents of the ChAD peptide and other 13-mer peptides evaluated in this study (peptide numbering is according to ref. [29])

In addition, we found that under the same experimental in vitro condition, purified *Mtb*-HigA1[ΔC42] mutant, which was shown to be fully soluble and functional in vivo in the absence of chaperone[29], was isolated as a dimer (Fig. 5a), as observed with known chaperone-independent HigA antitoxins[38,39]. SAXS was also used to characterize full-length *Mtb*-SecB[TA]/HigA1. The corresponding scattering curve is significantly distinct from those obtained for *Mtb*-SecB[TA] alone or in complex with the peptide and reflects a well folded protein/protein complex (Fig. 4c) of molecular weight $114.4 \pm 0.1$ and $120.0 \pm 0.3$ kDa as obtained from Guinier and concentration-independent methods, respectively, and with $R_g$ and Dmax values of respectively $3.61 \pm 0.02$ and $12.0 \pm 0.2$ nm. Finally, we performed HDX-MS on the *Mtb*-SecB[TA]/HigA1 complex. Strikingly, binding of the chaperone to *Mtb*-HigA1 client resulted in the protection of the same stretches of residues as for *Mtb*-SecB[TA]/ChAD with the addition of residues 48–51

and 59–66, which are located in the continuity of the peptide-binding pocket (Supplementary Fig. 7b). This observation suggests a significantly larger binding interface between the chaperone and full-length *Mtb*-HigA1 when compared to the short ChAD peptide. We also did not find any specific destabilization of *Mtb*-SecB[TA] upon complex formation with *Mtb*-HigA1, which is in agreement with the Kratky analysis of the SAXS data.

As observed for most type II DNA binding antitoxins, *Mtb*-HigA1 can act as a repressor of its own *Mtb*-*higBA1* operon[40]. In order to further investigate the activity of soluble dimeric *Mtb*-HigA1 bound to *Mtb*-SecB[TA] tetramers, we have used SEC-MALS, native MS, and ITC to monitor the interaction between the purified *Mtb*-SecB[TA]/HigA1 complex and dsDNA using a 38-bp palindromic sequence derived from the *higB* P2 promoter. First, we found that *Mtb*-SecB[TA] alone does not interact with the 38-bp dsDNA sequence. In contrast, SEC-MALS experiments

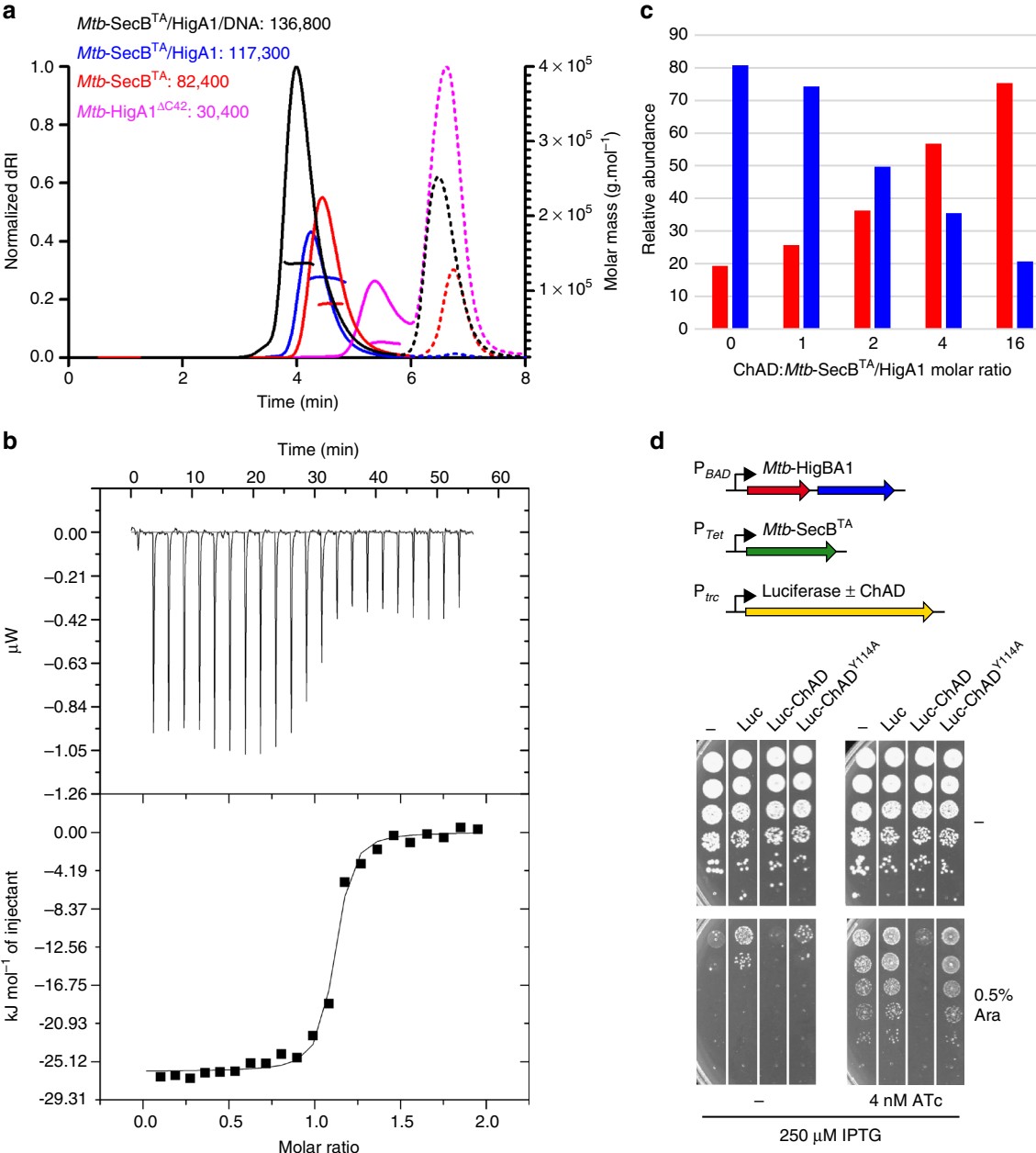

**Fig. 5** Characterization of full-length *Mtb*-SecB^TA/HigA1 chaperone/antitoxin complex. **a** SEC-MALS analysis. Continuous and dashed lines represent the variation of refractive index against elution time for *Mtb*-SecB^TA (red), *Mtb*-SecB^TA/HigA1 (blue), *Mtb*-HigA1^ΔC42 (violet), and *Mtb*-SecB^TA/HigA1/dsDNA (black) for protein and salt contributions, respectively. The experimentally measured molar mass distribution (same color code) and deduced mean molar mass (in g mol⁻¹) are indicated for each elution peak. Calculated molecular masses of proteins and protein complexes used for experiments (in daltons): *Mtb*-SecB^TA tetramer, 81,552; *Mtb*-SecB^TA/HigA1 hetero-hexamer, 115,068; *Mtb*-HigA1^ΔC42 dimer, 27,994; *Mtb*-SecB^TA/HigA1/dsDNA, 138,420. **b** ITC profile of 38-bp dsDNA titrated into *Mtb*-SecB^TA/HigA1 (upper and lower parts represent raw and integrated binding heats, respectively). **c** Native MS-derived relative abundance of *Mtb*-SecB^TA/HigA1 (blue bars) and *Mtb*-SecB^TA (red bars) obtained upon titration of *Mtb*-SecB^TA/HigA1 with 0, 1, 2, 4, 16 molar equivalents of ChAD. **d** Overexpression of luciferase-ChAD chimera activates *Mtb*-HigB1. *E. coli* WPtet57 strain expressing chromosomally encoded *Mtb*-SecB^TA under the control of an anhydrotetracycline inducible promoter was co-transformed with pC6-*Mtb*-HigBA1 and the pSE380-based vector (−), luciferase (Luc), or the Luc-*Mtb*-ChAD and Luc-*Mtb*-ChAD^Y114A chimeras. Transformants were grown to mid-log phase, serially diluted and spotted on LB ampicillin chloramphenicol agar plates without or with IPTG (to express Luc and Luc chimeras), arabinose (to express *Mtb*-HigBA1), and anhydrotetracycline (to express *Mtb*-SecB^TA) inducers as indicated. Plates were incubated overnight at 37 °C

demonstrated that *Mtb*-SecB^TA/HigA1 forms a single stable molecular species in the presence of DNA, most likely corresponding to 4:2:1 *Mtb*-SecB^TA/HigA1/DNA stoichiometry (Fig. 5a), with a shift in the hydrodynamic radius from 4.4 to 5.0 nm upon DNA binding. We verified by SEC-MALS that no interaction occurs between *Mtb*-SecB^TA/HigA1 and a nonspecific dsDNA such as a 16-bp oligonucleotide derived from the *RRM1*

promoter[41]. We also used native MS to analyze *Mtb*-SecB^TA/HigA1 in complex with its specific DNA target and observed that they indeed form a 4:2:1 complex of molecular weight 138.25 kDa (Supplementary Fig. 9a). In addition, ITC experiments showed that the *higB* P2 promoter DNA fragment interacts with the hexamer with a high affinity ($2.2 \pm 0.5 \times 10^7$ M⁻¹ at 25 °C) (Fig. 5b). The ITC-derived thermodynamic parameters indicated

an enthalpically and entropically favored binding ($\Delta H = -26.1$ kJ mol$^{-1}$ and $-T\Delta S = -15.9$ kJ mol$^{-1}$) suggesting association to the major groove of DNA[42]. The stoichiometry obtained by ITC for this specific association is 1.1, consistent with a binding involving one palindromic DNA sequence per *Mtb*-HigA1 dimer present in the *Mtb*-SecB$^{TA}$/HigA1 hetero-hexamer. Measuring the temperature dependence of the enthalpy gave a negative heat capacity of binding ($-1.3$ kJ mol$^{-1}$ K$^{-1}$) as typically reported for the majority of sequence-specific protein/DNA associations[43]. Together these results show that the highly unstable *Mtb*-HigA1 antitoxin can form a functional dimer following binding to its specific chaperone partner.

It has been proposed that disrupting the TA interfaces by small molecules or peptides to trigger endogenous toxin activation in vivo could represent a novel antibacterial tool to induce bacterial death from the inside[44]. The tripartite TAC system significantly differs from classical two-component TA systems due to the fact that the TAC antitoxin has acquired a C-terminal ChAD segment, which efficiently triggers antitoxin aggregation and/or degradation in the absence of chaperone, thus leading to toxin activation. Therefore, we hypothesized that disrupting an earlier step in the toxin inactivation pathway, namely the interaction between the chaperone and its antitoxin client, might lead to growth inhibition. To begin to approach this specific point, we first used native MS experiments to investigate whether a stably formed *Mtb*-SecB$^{TA}$/HigA1 complex could be disrupted in vitro by increasing concentrations of ChAD peptide. In the absence of peptide, we mainly found 4:2 *Mtb*-SecB$^{TA}$/HigA1 hetero-hexamers together with *Mtb*-SecB$^{TA}$ tetramers, with MW of 114.9 and 81.6 kDa, respectively (Supplementary Fig. 9b). Upon incubation with low amounts of ChAD, we observed peptide binding to both tetrameric *Mtb*-SecB$^{TA}$ (as described above) and to the *Mtb*-SecB$^{TA}$/HigA1 hexamer (Supplementary Fig. 9c, d). At higher peptide concentrations, more ChAD peptides bound to the hexamer, leading to the progressive dissociation of the tetramer from the complex (Fig. 5c and Supplementary Fig. 9e, f). These results suggest that a preformed antitoxin/chaperone complex in vivo might also be disrupted by the presence of a ChAD competitor, thus leading to a free active toxin and the subsequent inhibition of bacterial growth. To test such a hypothesis, we constructed luciferase chimeras containing a grafted C-terminal ChAD extension, either wild type or carrying the Y114A mutant unable to interact with the chaperone, overexpressed them in the presence of TAC and monitored bacterial growth in our toxicity rescue assay (Fig. 5d). Remarkably, we found that expression of the luciferase chimera with wild-type ChAD sequence could severely inhibit bacterial growth in the presence of TAC, while the Y114A mutant or the luciferase without ChAD could not. Together these results suggest that *Mtb*-SecB$^{TA}$/HigA1 complexes, with or without *Mtb*-HigB1 toxin, can be disrupted both in vitro and in vivo.

## Discussion

In this work, we provide biophysical and structural insights into the mechanism of chaperone addiction. We solved the high-resolution crystal structure of a TA-associated chaperone bound to the ChAD region of its antitoxin client. This structure reveals major differences in binding mode when compared to SecB–SecA or SecB-presecretory substrate complexes (Fig. 6) and identifies major contact surface residues involved in the chaperone–antitoxin interface.

The first structural report of a complex between SecB and a partner protein was the 2.8 Å X-ray structure of *Hi*-SecB with the C-terminal 27 residues of SecA (SecAc)[8]. In this structure, SecAc adopts a totally different conformation than the one adopted by

the *Mtb*-HigA1 (104–116) region and binding to the *Hi*-SecB chaperone occurs at a very different position with respect to the tetramer (Fig. 6 and Supplementary Fig. 1a). In this case, there are two bound SecAc peptides per *Hi*-SecB tetramer and each peptide interacts with the acidic cluster found on the eight-stranded β-sheet, resulting in a predominantly electrostatic interface[8] (Fig. 1b and Supplementary Fig. 3). More recently, a comprehensive NMR analysis of the interaction between *Ec*-SecB and two different secretory protein clients (namely PhoA and MBP) in the unfolded state has been published[6]. The solution structure of the different complexes shows how a single-protein client wraps around the *Ec*-SecB tetrameric assembly and is in agreement with earlier biochemical data showing that *Ec*-SecB can interact with long stretches of unfolded protein substrates[5,45]. Here, we mainly refer to the *Ec*-SecB–PhoA complex, which was the most thoroughly analyzed in the NMR study (Supplementary Fig. 1a). In this case, the client protein interacts through long unstructured hydrophobic segments that mostly fill the chaperone hydrophobic grooves (Fig. 1b and Supplementary Fig. 3). Remarkably, our data show that binding of PhoA precursor to *Ec*-SecB shares no apparent similarity with binding of ChAD to *Mtb*-SecB$^{TA}$. Indeed, we found that the ChAD peptide binding sites on *Mtb*-SecB$^{TA}$ run perpendicular with respect to the different PhoA binding sites in *Ec*-SecB (Supplementary Fig. 10) and that only few residues of *Ec*-SecB that interact with PhoA have counterparts in *Mtb*-SecB$^{TA}$ that interact with ChAD. Finally, although some of these residues of the contact surface spatially overlap, the interaction modes are significantly different (Fig. 1b). This is especially true for the three W108, R110, and Y114 residues of ChAD that are involved in strong interactions with *Mtb*-SecB$^{TA}$ (Supplementary Fig. 11).

Analysis of the full-length *Mtb*-SecB$^{TA}$/HigA1 complex shows that the aggregation-prone antitoxin of TAC forms a dimer when bound to the *Mtb*-SecB$^{TA}$ chaperone. The dimeric nature of the chaperone-bound antitoxin is in agreement with the observation that the chaperone-independent *Mtb*-HigA1 deleted of its C-terminal ChAD region (*Mtb*-HigA1$^{\Delta C42}$) also forms a soluble dimer and is functional in vivo in the absence of the chaperone (this work and ref. [29]). Such data strongly suggest that the chaperone-bound antitoxin dimer could reach a functional form. This is further supported by the fact that chaperone-bound *Mtb*-HigA1 has the ability to bind to its native promoter region in vitro and to inhibit the toxin in vivo. Whether such a conformation represents the fully folded antitoxin remains to be determined. Although SecB chaperones mainly assist protein export by preventing the folding of presecretory protein clients, our data show that some SecB chaperones can also assist the folding of cytosolic substrates such as the *Mtb*-HigA1 antitoxin. This property is in agreement with the observation that overexpressed *Ec*-SecB can partially rescue the bacterial growth defect of an *E. coli* strain lacking both major cytosolic chaperones DnaK and Trigger Factor, and that *Ec*-SecB can bind cytosolic proteins co- and/or post-translationally in vitro[46].

Our data also reveal that *Mtb*-SecB$^{TA}$ can potentially accommodate four ChAD peptides, although in this work we show that only one *Mtb*-HigA1 dimer is bound to one *Mtb*-SecB$^{TA}$ tetramer. Yet, the full-length ChAD region of *Mtb*-HigA1 contains a second stretch of residues ($_{137}$WARHISVR$_{144}$) that is similar to the first motif ($_{108}$WHRLSSYR$_{115}$) present in the ChAD peptide used in this work. Since these motifs resemble the previously identified SecB-binding motifs[4], that they both contain at least the key tryptophan and arginine residues involved in tight interaction with the chaperone, and that they are separated by a stretch of 21 amino-acid residues, such additional segment might well contribute to a secondary, lower affinity binding site. Note that in this case, binding would require a structural rearrangement at the

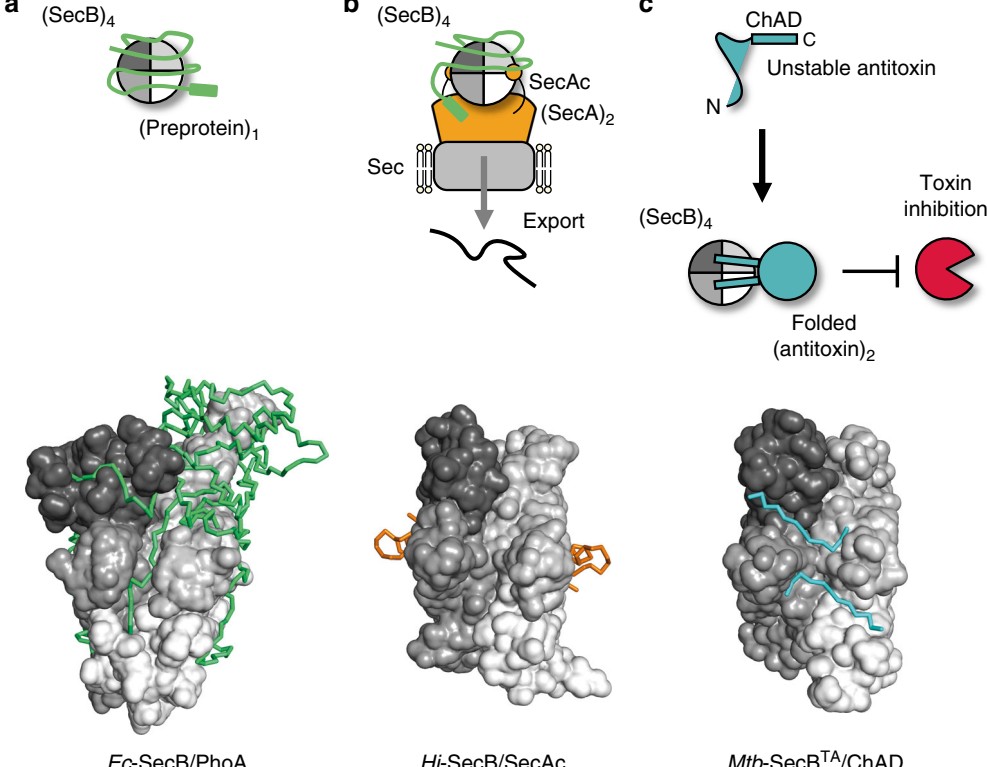

**Fig. 6** High structural plasticity of SecB proteins favors ligand avidity. Model showing the different SecB-binding modes based on NMR and X-ray data. **a** SecB displays robust antifolding activity when bound to presecretory protein clients as illustrated by the *Ec*-SecB/PhoA structure, **b** SecB binds to its SecA partner to perform protein export as illustrated by the *Hi*-SecB/SecAc structure, and **c** SecB associated with TA systems binds to the chaperone addiction region of the antitoxin (ChAD), facilitating its folding and subsequent toxin inhibition. The oligomeric states of the different proteins are indicated in parentheses

chaperone level to unlock steric constraints due to its distorted dihedral symmetry. In addition, secondary binding events could further increase the affinity of the antitoxin for the chaperone, when compared to the peptide alone. On the other hand, these two binding sites might stay readily accessible to other substrates or partners, including SecA or presecretory proteins, suggesting that they could also contribute to the activation cycle of the TAC system in response to certain stress, as previously proposed[1].

## Methods

**Bacterial strains and culture conditions.** *E. coli* strains W3110 and W3110 *secB*::Cm[R35] have been described. Bacteria were routinely grown in LB medium supplemented when necessary with kanamycin (50 μg ml$^{-1}$), chloramphenicol (15 μg ml$^{-1}$), or ampicillin (50 μg ml$^{-1}$). For strain W3110 Δ*araBAD*::P$_{Tet}$-*Rv1957* (Kan[R]) Δ*secB*, named WPtet57, the chromosomal *araBAD* operon was replaced by a cassette containing the *Mtb*-SecB[TA] under the control of an anhydrotetracycline inducible promoter using recombineering technology[47].

**Plasmid constructs.** Plasmids pMPMK6[48], p29SEN[49], p29-*Mtb*-SecB[TA30], pET-Duet-$_{6HIS}$*Mtb*-SecB[TA]-HigA1, pET15b-*Mtb*-SecB[TA] and pK6-*Mtb*-HigBA1[22], and pET15b (Novagen) have been described. The p29-*Mtb*-SecB[TA] mutant derivatives were constructed by QuickChange mutagenesis (Stratagene) using appropriate primers (Supplementary Table 3) and p29-*Mtb*-SecB[TA] as DNA template. The constructs were sequenced verified using appropriate primers. To construct pET15b-*Mtb*-HigA1$^{\Delta C42}$, *Mtb*-*higA1*$^{\Delta C42}$ was PCR amplified with primers pBAD FOR(5′-CGCAACTCTCTACTGTTTC-3′) and *higA* ΔC42 REV H/B (5′-TTTAA GCTTGGATCCTCACGTAGGCACTTCGCGAAGCGTG-3′) and cloned into a NdeI-BamHI fragment into pET15b vector. To construct plasmids pSE-Luc and pSE-Luc-*Mtb*-ChAD, the EcoRI/HindIII digested fragments containing the *Photinus pyralis* luciferase gene, either WT or fused to the *Mtb*-HigA1 (ChAD) C-terminal region[22], were subcloned into EcoRI/HindIII digested pSE380ΔNcoI[49]. Plasmid pSE-Luc-*Mtb*-ChAD was used as template to construct plasmid pSE-Luc-*Mtb*-ChAD$^{Y114A}$ by QuickChange mutagenesis with *higA1* ChAD codon change TAT→GCC. To construct plasmid pC6-*Mtb*-*higBA1*, the EcoRV-SphI digested

*Mtb*-*higBA1* fragment from pK6-*Mtb*-*higBA1* was cloned into the chloramphenicol resistant plasmid pBAD33[50] digested with the same enzymes.

**In vivo TA control by chaperones.** Toxin inhibition assay in vivo was performed as described[22], except that *Mtb*-SecB[TA] and its mutant derivatives were cloned on the low-copy plasmid p29SEN[49]. Briefly, fresh transformants of *E. coli* strains co-expressing the p29SEN-based chaperone and the pMPMK6-based *Mtb*-HigBA1 pair under the control of the P$_{trc}$ and P$_{BAD}$ promoter, respectively, were grown in LB ampicillin kanamycin to mid-log phase, serially diluted and spotted on LB ampicillin kanamycin agar plates with or without IPTG (to induce the chaperone) and arabinose (to induce *Mtb*-HigBA1) as indicated in the figure legends. Plates were incubated overnight at 37 °C. When strain WPtet57 was used, 4 nM of anhydrotetracycline was added to induce expression of *Mtb*-SecB[TA].

**In vivo chaperone assays.** It was previously shown that *E. coli secB* mutants do not grow at low temperature and are sensitive to novobiocin antibiotic[30,34,35], likely due to the intrinsic cold sensitive property of the Sec translocon and to an impaired outer membrane[1,51]. In this case, it was found that growth at low temperature and resistance to novobiocin requires a functional interaction between SecB and SecA in vivo[30,35]. Complementation of SecB chaperone activity in vivo at 18 °C was performed as follows. Fresh transformants of W3110 Δ*secB*::CmR containing p29-*Mtb*-SecB[TA] or its mutant derivatives were grown at 37 °C to mid-log phase in LB ampicillin, serially diluted, and spotted on LB ampicillin agar plates in the absence or presence of IPTG inducer. Plates were incubated at 37 °C overnight or for 3 days at 18 °C. For novobiocin resistance, mid-log phase cultures of strain W3110 Δ*secB* containing p29-*Mtb*-SecB[TA] or its mutant derivatives in LB ampicillin were serially diluted and spotted on LB ampicillin agar plate with 30 μg ml$^{-1}$ novobiocin, and incubated overnight at 37 °C.

**Protein purification.** *Mtb*-SecB[TA] with a N-terminal His$_6$-tag was purified following overexpression in a BL21(λDE3) *E. coli* strain harboring plasmid pET15b-Rv1957 as described[22]. The *Mtb*-SecB[TA]/HigA1 complex was overexpressed from plasmid pDUET-1 transformed in BL21(λDE3) giving his-tagged *Mtb*-SecB[TA] and untagged HigA1. For production, cells (500 ml) were grown at 30 °C to an OD$_{600}$ of 0.5 in LB medium containing 100 μg ml$^{-1}$ ampicillin, and expression of the protein or the complex was induced with 1 mM IPTG for 3 h. Cells were harvested, washed

in phosphate-buffered saline (PBS) buffer (pH 7.2), and stored at −80 °C. Cells were thawed in lysis buffer (20 mM Tris, 2 mM EDTA, 2 mM DTT, 20 mM imidazole, 100 mM NaCl, pH 7.5) containing 0.5 mg ml⁻¹ lysozyme, sonicated on ice using five intermittent pulses of 30 s, and centrifuged at 30,000 *g* for 45 min at 4 °C. The supernatant was loaded onto a 1-ml HiTrap HP (GE Healthcare) FPLC column, pre-equilibrated in buffer A (20 mM Tris, 0.1 mM EDTA, 200 mM NaCl, pH 7.5) supplemented with 20 mM imidazole. The column was washed with buffer A supplemented with 40 mM imidazole. Elution was achieved with buffer A supplemented with 160 mM imidazole and the more concentrated fractions were pooled. The pooled fractions containing *Mtb*-SecB$^{TA}$ (approximately 20 mg) or *Mtb*-SecB$^{TA}$/HigA1 (approximately 10 mg) were applied onto a HiLoad 16/60 Superdex 200 prep-grade column (GE Healthcare) pre-equilibrated with buffer GF (20 mM MES, 200 mM NaCl, pH 6.5). *Mtb*-SecB$^{TA}$ eluted as two major peaks characterized as octamer and tetramer using SEC-MALS experiments. Only the fractions corresponding to the tetramer, which eluted at a volume of 72.5 ml, were pooled and subjected to thrombin cleavage to remove the N-terminal His$_6$-tag. *Mtb*-SecB$^{TA}$/HigA1 eluted as a major peak at 68.2 ml, which was further characterized as a hetero-hexamer by SEC-MALS. Removal of the N-terminal His$_6$-tag on *Mtb*-SecB$^{TA}$ was performed by incubating the protein at 0.5 mg ml⁻¹ in GF buffer with thrombin under shaking for 3 h at room temperature. After checking by sodium dodecyl sulphate polyacrylamide gel electrophoresis (SDS-PAGE), the cleaved sample was loaded on a HiTrap HP column pre-equilibrated in buffer A supplemented with 20 mM imidazole to remove the tagged protein. Further purification by gel filtration in the conditions described above led to a single peak. Finally, gel filtrated proteins were concentrated with Vivaspin 20 (10 kDa and 30 kDa cutoff for *Mtb*-SecB$^{TA}$ and *Mtb*-SecB$^{TA}$/HigA1, respectively) and either freshly used, for crystallization, or flash-frozen in liquid nitrogen and stored at −20 °C while awaiting biophysical characterization.

*Mtb*-HigA1$^{\Delta C42}$ (*Mtb*-HigA1 deleted of the last 42 residues) was overexpressed from pET15b-HigA1$^{\Delta C42}$ in BL21(λDE3) as a fusion protein containing a N-terminal His$_6$-tag. For production, cells (2 l) were grown to an OD$_{600}$ of 0.6–0.7 in LB medium containing 100 µg ml⁻¹ ampicillin at 30 °C, and expression of the *Mtb*-HigA1$^{\Delta C42}$ was induced with 1 mM IPTG for 4 h. Cells were harvested, washed in PBS buffer (pH 7.2), and stored at −80 °C. It is noteworthy that all buffers used for purification of *Mtb*-HigA1$^{\Delta C42}$ contained 0.2 mM AEBSF. Cells were thawed in lysis buffer containing lysozyme at 0.5 mg ml⁻¹, sonicated on ice using five intermittent pulses of 30 s, and centrifuged at 30,000 *g* for 45 min at 4 °C. The supernatant was loaded onto a 1-ml HiTrap HP (GE Healthcare) FPLC column pre-equilibrated in buffer A supplemented with 20 mM imidazole. Washing and elution of the column were achieved with buffer A supplemented with 40 and 300 mM imidazole, respectively. *Mtb*-HigA1$^{\Delta C42}$-containing fractions were pooled and concentrated on a Vivaspin 20 (10 kDa) and the recovered retentate applied onto a Superdex 75 10/300 GL column (GE Healthcare) pre-equilibrated with buffer GF. Fractions containing *Mtb*-HigA1$^{\Delta C42}$ appeared as a continuous elution (ranging from 11 to 27 ml) confirmed using SDS-PAGE gel. Eluted fractions were pooled, concentrated, flash-frozen and stored at −20 °C while awaiting biophysical characterization.

Concentrations were determined by measuring UV absorption at 280 nm with a NanoDrop 2000 spectrophotometer (ThermoScientific) and using calculated molar extinction coefficients (in M⁻¹ cm⁻¹) of 15,930 for untagged *Mtb*-SecB$^{TA}$ (20,388 Da), 94,460 for untagged *Mtb*-SecB$^{TA}$/HigA1 (considered as a hetero-hexamer of 115,068 Da in all calculations), and 2980 for tagged *Mtb*-HigA1$^{\Delta C42}$ (13,997 Da).

**Crystallization and structure determination.** The 13-mer ChAD peptide corresponding to residues $_{104}$EVPTWHRLSSYRG$_{116}$ of *Mtb*-HigA1 was synthesized by Schafer-N (Copenhagen, Denmark) to >95% purity and solubilized in GF buffer. Purified *Mtb*-SecB$^{TA}$ was concentrated to 30 mg ml⁻¹ (1.5 mM) in GF buffer and mixtures of *Mtb*-SecB$^{TA}$ and ChAD peptide, supplemented or not with 5% of glycerol, were subjected to crystallization trials performed at 285 K using the vapor diffusion method in sitting drops. Final peptide concentrations were chosen to reach a peptide/protein molar ratio of 3.0 starting from *Mtb*-SecB$^{TA}$ protein and ChAD peptide (MW = 1588 Da and calculated molar extinction coefficient of 6990 M⁻¹ cm⁻¹) stock solutions at 1.5 and 18.5 mM, respectively. Drops consisting of 200 nL of complex and 200 nL of precipitant solution were equilibrated against 80 µL of reservoir solution. Several crystallization conditions were obtained in the presence of PEG of various molecular weights and additives such as calcium acetate, MgCl$_2$, or LiCl. The best diffracting crystals were grown with the protein in GF buffer supplemented with 5% glycerol and in the presence of 100 mM Tris, 25–27% PEG 1500, 80–240 mM calcium acetate, 10% DMSO, pH 8.0. They were directly frozen under a cryogenic nitrogen stream and stored in liquid nitrogen before data collection at 100 K. Diffraction data were measured from tetragonal crystals diffracting to ca. 1.8 Å at the ESRF synchrotron (Grenoble, France) on beamline ID29 using a PILATUS 6M detector. X-ray images were indexed, integrated and scaled on-line using autoP-ROC[52]. Programs from the CCP4 software suite[53] were used for subsequent crystallographic calculations[54]. The structure was determined by the molecular replacement method with PHASER[55] and a starting model derived from the structure of SecB from *H. influenzae*[31] (PDB code 1FX3). An initial model was automatically built with Buccaneer[56] and used as a starting point for iterative refinement with Refmac[57] and manual rebuilding using Coot[58]. Final refinement led to $R_{work}/R_{free}$ values of 0.212/0.265. Data collection and refinement statistics are given in Table 1.

Quality of the structure was validated during the PDB deposition process. Protein structure databases were searched and structures were superimposed with Dali software[59,60] and SUPERPOSE from the CCP4 software package. Molecular interfaces were explored using PISA calculations as implemented on the EMBL-EBI server[61]. All structures and the electrostatic potential of protein surfaces were visualized with PyMOL[62]. The sequence alignment was generated with ESPript 3[63].

**MD calculations.** The X-ray structure was used to generate a starting template of the *Mtb*-SecB$^{TA}$/ChAD complex after building missing residues of the chaperone (Table 1), finally accounting for 161 residues from Asp6 to Met166. The tleap module of Amber14[64] was used to generate a periodic octahedral box containing the complex extended by 10 Å and filled with TIP3P water molecules and sodium ions to neutralize the overall charge of the system. Energy minimization and MD simulation were performed with the parallel version of the pmemd module of Amber14. After an initial energy minimization with progressively reduced restraints on the position of all atoms (excluding solvent), 100 ps MD were run with temperature varying linearly from 0 to 300 K at constant volume, followed by a 400 ps MD at 300 K and a constant pressure of 1 bar, with the atomic coordinates saved every 2 ps. Three MD simulations of 20 ns were performed at a constant pressure of 1 bar, with atomic coordinates saved every 10 ps. The corresponding trajectories were concatenated in order to build a 60 ns MD simulation. Distance and angle variations in the course of the simulation were analyzed with the cpptraj module[65] from Amber14.

**Size-exclusion chromatography and multi-angle light scattering.** Samples of *Mtb*-SecB$^{TA}$ at 49.1 µM (1.0 mg ml⁻¹), *Mtb*-SecB$^{TA}$/HigA1 at 6.5 µM (0.75 mg ml⁻¹), and *Mtb*-HigA1$^{\Delta C42}$ at 100 µM (1.4 mg ml⁻¹) in 20 mM MES, 200 mM NaCl, pH 6.5 were analyzed on a Superdex 200 Increase 5/150 (GE Healthcare) with multi-angle light scattering. The column was equilibrated with the same buffer, previously sterilized on a 0.1-µm filter, on an Agilent 1260 Infinity LC chromatographic system (Agilent Technology). Data were collected on a DAWN HELEOS-II 8-angle and Optilab T-rEX refractive index detector (Wyatt Technology, Toulouse, France). We loaded 75–100 µl of each protein sample and separation was performed at a flow rate of 0.4 ml min⁻¹ at 15 °C. Results were analyzed with ASTRA 6.0.2.9 software (Wyatt Technology Corp.).

**SAXS experiments.** SAXS experiments were conducted in HPLC mode on the BM29 beamline at the ESRF synchrotron, Grenoble, France. *Mtb*-SecB$^{TA}$ (40 µl at 474 µM and 50 µl at 879 µM), *Mtb*-SecB$^{TA}$/HigA1 (60 µl at 136.5 µM) samples in 20 mM MES, 200 mM NaCl, pH 6.5 were injected onto a Superdex 200 Increase 5/150 column, equilibrated with the same buffer supplemented with 2 mM DTT, and run at 0.3 ml min⁻¹ and a temperature of 15 °C. *Mtb*-SecB$^{TA}$ was also measured in the presence of the ChAD peptide at a protein:peptide molar ratio of 6:4 (50 µl at 881 µM protein and 588 µM peptide), and the column was equilibrated in the same buffer supplemented with 200 µM of ChAD peptide. Data were collected in a Q range of 0.035–4.9 nm⁻¹, where $Q = 4\pi\sin\theta/\lambda$, with a scattering angle of $2\theta$. A total of 780 frames of 1 ms each were collected throughout the whole elution time. Data, displayed on an absolute scale using water calibration, were processed with the ATSAS suite[66]. After buffer subtraction, Guinier analysis at low Q was performed on each curve using the PRIMUS program to check for the stability of the associated radius of gyration, and the selected frames were averaged. Molecular weights were determined using the concentration-independent method[36] and Guinier analysis using PRIMUS within the limit $Q.R_g < 1$. Regularized indirect transforms of the scattering data were performed with the program GNOM to obtain $P(r)$ functions of interatomic distances.

**Microscale thermophoresis.** MST assays were carried out on a NanoTemper Monolith NT.115 instrument (NanoTemper Technologies GmbH, Germany). Briefly, *Mtb*-SecB$^{TA}$ at a concentration of 7.5 µM was labeled with the RED fluorescence dye NT-647 according to manufacturer's protocol in 20 mM MES, 200 mM NaCl, 5% DMSO, pH 6.5. Labeling and removal of free dye were performed within 45 min. Subsequently, 100 nM of NT-647-labeled protein was mixed with various concentrations of ChAD peptide (3 nM to 100 µM) in 20 mM MES, 200 mM NaCl, 0.025% Tween 20, pH 6.5. After incubation for 5 min at room temperature, 20 µl of the mixtures were loaded into premium coated capillaries and MST data were collected at 22 °C under 40% infrared laser power and 50% light-emitting diode power. NanoTemper analysis software (2.1.2) was used to fit the independent experimental data and determine dissociation constants ($K_d$) and associated errors.

**Isothermal titration calorimetry.** ITC experiments were carried out at 12 °C on a MicroCal ITC200 system (Malvern Instruments) with purified *Mtb*-SecB$^{TA}$ and ChAD peptide in 20 mM MES, 200 mM NaCl, pH 6.5. The titration protocol comprised a preliminary 0.4 µl injection followed by 19 (resp. 39) consecutive injections of 2 µl (resp. 1 µl) of peptide solution into the sample cell containing the protein. Spacing between each injection was 180 s. Control titrations of peptide solution into the protein-free experimental buffer were also performed. Data were analyzed with MicroCal Origin 5.0 software on the basis of a "one set of sites" which gave the best fit to experimental data. Thermodynamic parameters and errors were obtained from four measurement repeats with concentrations in the range 18–48 µM and 80–620 µM for the protein and the peptide, respectively.

**DNA binding assays using SEC-MALS and ITC.** The 38-bp specific DNA duplex used for SEC-MALS and ITC (5′-TAGCTGGGCATATAGGTTACAGCCTA-TATTCTGGTATA-3′ and its complementary strand) was synthesized and HPLC-purified by Sigma-Aldrich. The two complementary oligonucleotides were solubilized in water and hybridized by heating to 98 °C followed by cooling, and the resulting DNA duplex was then lyophilized (Alpha 1–2 LDplus, Christ). The powder was then suspended in 20 mM MES, 200 mM NaCl, pH 6.5 at a concentration of 800 μM. For SEC-MALS experiments, the 38-bp dsDNA oligomer (190 μM) was mixed with purified *Mtb*-SecB^TA/HigA1 (23.5 μM) in the same buffer, at a DNA/protein molar ratio of 1.28, followed by a 5-min incubation at 4 °C. The DNA–protein sample (51 μg) was then injected at 18 °C. For ITC experiments, the 38-bp dsDNA oligomer (170–190 μM) was used for titration into purified *Mtb*-SecB^TA/HigA1 (18–23.5 μM) at 12, 18, and 25 °C. Both SEC-MALS and ITC experiments were conducted using the same experimental conditions as described above.

**Native MS.** Prior to native MS analysis, *Mtb*-SecB^TA, *Mtb*-SecB^TA/HigA1, and *Mtb*-SecB^TA/HigA1/DNA samples were desalted in 200 mM ammonium acetate, pH 7 using Micro Bio-Spin devices (Bio-Rad, Marnes-la-Coquette, France) at molar concentrations of 11.8 μM (tetramer), 9.7 (hexamer), and 10.5 μM (heptamer), respectively. ChAD peptide stock solutions were prepared in the same buffer at a final concentration of 10.9 mM. ChAD peptide/*Mtb*-SecB^TA molar ratios were from 0 to 20, corresponding to peptide concentrations ranging from 0 to 195 μM and a maximum of 20 peptide molecules per chaperone tetramer. For the interaction with *Mtb*-SecB^TA/HigA1, molar ratios were chosen to cover a range from 0 to 16 peptide molecules per hexamer. The 13-mer C2, C3, and C7 peptides (see ref. [29]) used as controls were prepared from concentrated stock solutions in water and diluted 100-fold in the ammonium acetate buffer to a final concentration of 112.5 μM and injected in the presence of *Mtb*-SecB^TA at a ratio of 8 peptides per tetramer. All peptides were added to the chaperone or the chaperone–antitoxin complex right before injection. Samples were analyzed on a SYNAPT G2-Si mass spectrometer (Waters, Manchester, UK) running in positive ion mode and coupled to an automated chip-based nano-electrospray source (Triversa Nanomate, Advion Biosciences, Ithaca, NY, USA). The voltage applied to the chip and the cone voltage were set to 1.6 kV and 150 V, respectively. The instrument was calibrated with a 2 mg ml⁻¹ cesium iodide solution in 50% isopropanol. Raw data were acquired with MassLynx 4.1 (Waters, Manchester, UK) and deconvoluted with UniDec[67] using the following parameters: $m/z$ range: 4000–8000 Th; subtract curved: 5; Gaussian smoothing: 10; bin every 10 Th; charge range: 10–30; mass range: 80,000–125,000 Da; sample mass: every 10 Da; peak FWHM: 10 Th; peak detection range: 250 Da, and peak detection threshold: 0.02–0.1. $K_d$ values were obtained by fitting titration curves in the data collector utility from Unidec with the following parameters: "How to extract": Local Max, "Number of proteins": 1 (tetrameric concentration), "Number of Ligands": 4, "Protein and Ligand Models": All KD's Free, "Binding Sites Per Proteins": 4.

**Hydrogen–deuterium exchange coupled to MS.** Automated Hydrogen–Deuterium eXchange coupled to MS (HDX-MS) experiments were performed on a Synapt G2-Si mass spectrometer (Waters Scientific, Manchester, UK) coupled to a Twin HTS PAL dispensing and labeling robot (LEAP Technologies, Carborro, NC, USA) via a NanoAcquity system with HDX technology (Waters, Manchester, UK). Briefly, 5.2 μL of protein at 15 μM were diluted in 98.8 μL of protonated (for peptide mapping) or deuterated buffer (20 mM MES pD 6.5, 200 mM NaCl) and incubated at 20 °C for 0, 0.5, 5, 10, and 30 min. Totally, 99 μL were then transferred to vials containing 11 μL of pre-cooled quenching solution (500 mM glycine at pH 2.3). After 30 s quench, 105 μL were injected to a 100 μL loop. Proteins were digested on-line with a 2.1 mm × 30 mm Enzymate^TM BEH Pepsin column (Waters Scientific, Manchester, UK). Peptides were desalted for 2 min on a C18 pre-column (Acquity UPLC BEH 1.7 μm, VANGUARD) and separated on a C18 column (Acquity UPLC BEH 1.7 μm, 1.0 × 100 mm) by a linear gradient (2–40% acetonitrile in 13 min). Experiments were run in duplicates and protonated buffer was injected between each duplicate to wash the column and avoid cross-over contamination. Starting concentrations were 35 μM for *Mtb*-SecB^TA alone and in the presence of 6 molar equivalent of ChAD peptide (280 μM) and 18 μM for *Mtb*-SecB^TA/HigA1. Peptide identification was performed with ProteinLynx Global SERVER (PLGS, Waters, Manchester, UK) based on the MSE data acquired on the nondeuterated samples. Peptides were filtered in DynamX 3.0 with the following parameters: peptides identified in at least 3 out of 5 acquisitions, 0.3 fragments per amino-acid, intensity threshold 1000.

## Data availability

The atomic coordinates and structure factors have been deposited in the Protein Data Bank, Research Collaboratory for Structural Bioinformatics, Rutgers University, New Brunswick, NJ (code 5MTW). The HDX-MS and native MS data are available as files Supplementary Data 1 and Supplementary Data 2, respectively. The source data underlying Figs. 4b, d, e and 5c are provided as a Source Data file. Other data that support the findings of this study are available from the corresponding authors on request.

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

## Acknowledgments

We thank Dounia El-Mazouni, Sevan Arabaciyan, Sylviane Julien, and Jean Lesne for technical help. We thank Marie-Pierre Castanié-Cornet for strain gift. We also thank the scientific staff of the European Synchrotron Radiation Facility (Grenoble, France), SOLEIL (Gif sur Yvette, France), and ALBA (Barcelona, Spain) for the use of their excellent data collection facilities. We particularly thank the staff of beamlines ID29 and BM29 at the European Synchrotron Radiation Facility, where the crystallographic and SAXS experiments were, respectively, conducted. This work was supported by grants SNF CRSII3_160703 to P.G. and ANR-13-BSV8-0010-01 to P.G. and L.M. The SEC-MALS, ITC and macromolecular crystallography equipment used in this study are part of the Integrated Screening Platform of Toulouse (PICT, IPBS, IBiSA). We thank the Fédération de Recherche Agrobiosciences, Interactions et Biodiversité (FR 3450, Casta-net-Tolosan, France) for making the MST equipment available to us. Mass spectrometry experiments were supported by the French Ministry of Research (Investissements d'Avenir Program, Proteomics French Infrastructure, ANR-10-INBS-08 to Odile Burlet-Schiltz) and the Région Midi-Pyrénées (Odile Burlet-Schiltz).

## Author contributions

V.Gu., P.B., P.G. and L.M. conceived and designed the research; V.Gu., P.B., J.M., V.G., A.J.S., S.D.R., N.S., A.M.C., L.Ma., I.M.M. and L.M. performed the experiments; V.Gu., P.B., C.B., J.M., V.G., L.Ma., P.G. and L.M. analyzed the data; P.G. and L.M. wrote the paper, with contributions from all the authors; P.G. and L.M. supervised the project.
