## [Peer Review File · Nature Communications]

Reviewers' comments:

Reviewer #1 (Remarks to the Author):

The manuscript of Guillet and colleagues describes some structural insights into the chaperone addiction of the *M. tuberculosis* HigBA. In the first part, the authors provide a thorough structural study of the Mtb SecB tetramer in complex with the ChAD peptide of the HigA1 antitoxin. The second part describes their various biochemical attempts at deciphering the architecture of the Mtb SecB/HigA1 complex. Whereas the structural part is conclusive, the biochemical part is rather ambiguous and might require further reconsideration. Thus, I have major concerns on the quality of the manuscript.

Major concerns:

Each of the different experiments (structural investigations, ITC, native MS using the ChAD peptides, and SEC-MALS) give different stoichiometry of HigA1 or ChAD peptide bound to tetrameric SecB.

1) The authors argue that in the crystal structure steric clashes prevent a fourth peptide molecule binding to SecB. It remains unclear whether this is simply due to crystal contacts or to the distorted D2 symmetry. Along this line, it's also unclear why the authors do not see a Calcium ion bound to chain C, although a peptide is bound. To which extent are these results biased by crystal artifacts?

2) ITC experiment Figure 4a. The design of the experiment in its current form does not allow to unambiguously determine the molar ratio of ChAD peptide and SecB. One might consider to vary the ratio ChAD to SecB in such experiments.

3) Furthermore, if the ITC experiment is correct (2 binding sites), why do the authors observe 3 peptide molecules bound in the crystal structure?

4) Why do the authors observe 4 ChAD peptides binding to SecB in native MS experiments? This is again in strong contrast to all previous results.

5) Eventually, once the authors use full-length HigA1 they observe a 4:2 ratio of SecB to HigA1. Which of the 3 (4?) binding sites are occupied?

The molecular biology experiments on mutated SecB expressed in *E. coli* are difficult to understand. It remains unclear to the reviewer to which extent the TAC part, the chaperone activity of SecB, or the expression rate and protein stability are causing the observed differences.

Minor comments:

The authors should show somewhere in the manuscript unbiased electron density maps for all 3 peptides.

In some aspects it would be helpful to the reader if figures would directly indicate which protein is expressed by which inducer.

The comparison of Mtb SecB with related pdb entries is lengthy and the authors might consider shortening this part.

Reviewer #2 (Remarks to the Author):

In their study Guillet and colleagues resolved the structure of the *Mycobacterium tuberculosis* SecB homolog bound to the HigA1 antitoxin C-terminal peptide ChAD. Type II toxin-antitoxin (TA) systems are bicistronic operons encoding a toxic protein that block crucial processes in the bacterial cell (i.e. translation, DNA replication), and an antitoxin that neutralizes its poisonous partner. Mycobacterial HigBA1 is a special type II TA where there exists a third component, the chaperone SecB, that assists the folding of the antitoxin HigA1 and allows the proper neutralization of the toxin HigB1. Yet this process was previously described, the structural details of interaction between SecB and HigA1 were unknown. In this study the authors resolved the crystal structure of SecB-ChAD to further understand how SecB chaperones can discern between substrates. The structure of SecB-ChAD shows major differences on binding compared to other SecB substrates (e.g. SecB-SecA). Guillet and colleagues identified key residues of SecB necessary to bind the antitoxin C-terminal peptide ChAD. It is also suggested that HigA1 antitoxin can reach an active form while bound to SecB. A large variety of techniques are used to validate these findings.

I personally think that the work presented by Guillet and colleagues has quality enough to be published in this journal. Authors use a large variety of techniques to validate the crystal structure of Mtb-SecB/ChAD. Also, the manuscript is clearly written and the data well presented. I only have a few questions and some comments that I think might clarify some parts of the text / Figures; I list them below:

1- "In absence of stress, the antitoxin-bound toxin is inactive and the bacteria grow normally."

Although it is true that TA systems have been associated to different stress conditions, including antibiotic exposure, heat shock or pH, TA systems are also widely distributed in plasmids, where they mediate plasmid stability. In this case there is no stress per se, but it is the lost of the plasmid the event that impairs the synthesis of new antitoxin and therefore the activation of the toxin. I think this concept has to be introduced at this point of the introduction for readers not familiarized with the topic. Otherwise a biased impression of the TA systems is transmitted.

2- "Accordingly, such a reversible control of bacterial growth by toxins has been involved in bacterial

adaptation to stress, persistence and virulence^{13,14}." Change reference 14 by Lobato-Márquez et al., 2016, FEMS Microbiol Rev, where a comprehensive revision of TA modules and its connection to virulence is provided.

3- "We also noted that the presence of calcium ions was important for the crystallization of the complex." What is the relevance of Ca²⁺ ions on SecB structure? What is the role of DMSO molecules on protein stabilization? Are they also important in SecB structure alone?

4- In the Results paragraph where the differences between the Mtb-SecBTA tertiary structure and its SecB counterparts are described, do the authors know how important are these regions for Mtb-SecBTA function? Have authors tried to create chimeras of *E. coli* SecB and Mtb-SecBTA for example, and test them in toxicity-neutralization or SecB chaperone generic ability assays?

5- In the Results Mtb-SecBTA / ChAD interaction section, do the authors know if the binding of peptides E, F and G occur at same time? Could it be a cooperative binding where interaction of peptide E increases the affinity for peptide F and binding of F for G?

6- "All the Mtb-SecBTA mutations that most significantly impaired TA control, with the exception of Y49A, additionally affected its ability to..." I do not understand why authors focus so much in the mutant Y49A when other mutants, such as D27A or F67A have similar neutralization ability in experiments shown in Fig.3.

7- How authors explain the gain of function of SecB mutants in chaperone generic ability? Is it due

to a change of substrate specificity?

8- Please indicate in Results section what ITC and MST stand for the first time these two techniques are mentioned in the text.

9- When authors analyzed the quaternary complexes between and Mtb-SecBTA and HigA1 by SEC-MALS and Mass-spec they found that SecB tetramer interacts with a HigA1 dimer. I might have missed something but, how is this result compatible with the crystal structure shown in Fig.1A where SecB was found to interact with 3 ChAD peptides? Is this an artifact of the crystals and SecB can only bind 2 ChAD peptides and therefore 2 antitoxin molecules? Have authors tried different buffer concentration (e.g. increasing ionic strength) to break possible unspecific interactions? I do not obviously find this issue discussed anywhere in the text.

10- In TA systems the binding of the antitoxin to the DNA promoter operator regions is modulated by the toxin:antitoxin ratios. How does the toxin HigB1 influence the DNA-binding capacity of the complex SecB-HigA1? How strong is the binding of HigA1 alone (in the absence of Mtb-SecB) to DNA?

11- In the Material and Methods section "In vivo TA control by chaperones" authors should indicate which inducers induce which gene. Fig.3 can be confusing.

12- Figure 3a: How many times were these spot assays done? This is not mentioned anywhere. Please indicate what genes are induced with each inducer in the Figure legend in the same way that is mentioned in Figure 5d.

Also, I would suggest adding lines close to the inducer concentrations indicating which set of strains were spotted under each inducer (Same as Fig. 5d).

13- Figure 3b: I think changing the color of SecB to a lighter grey would allow distinguishing the residue numbers more easily.

14- Figure 6: To be consistent adapt the color of the SecB interacting protein in the Hi-SecB/SecAc upper and bottom panels as done for Ec-SecB/PhoA and Mtb-SecB/ChAD.

15- Supplementary Figure 2: to make this figure cleared authors should consider putting the names of the Sec complexes in the Figure.

16- Supplementary Figure 3: It would be helpful if authors explained the basis for the novobiocin assay either on the Results or Material and Methods section. Although this is referred to other papers, it is not explain there either.

17- Supplementary Figure 5: authors should consider changing the order of the graphs so "a" is followed by "b" and "c" horizontally at the top and "d" and "e" horizontally at the bottom.

18- Have the authors tested the specificity of Ec-SecB or Hi-SecB to ChAD?

19- Following question 18, authors describe mutations that affect Mtb-SecB binding to HigA1 but increase the general chaperone function of Mtb-SecB. Have authors tried adapting those equivalent residues of Ec-SecB or Hi-SecB and changing them into the sequence of Mtb-SecB? By doing that they might see an increase in Ec-SecB or Hi-SecB binding to ChAD. This kind of experiments might elucidate extra information about SecB chaperones specificity for their partners.

20- In most type II TA systems, the antitoxin C-terminal domain is normally unstructured, and only gets structured when the antitoxin binds the toxin. I was wondering if in TAC systems the chaperone binding domains are somehow connected to the "classic" unstructured C-terminal

domain. Could it be an evolution of TA systems? Authors might comment this in the Discussion section.

21- Until recently it was believed that the main regulation of antitoxin levels in type II TAs occurred at the transcriptional level and at the protein level via protease (e.g. Lon, Clp) degradation. Here, the presence of a third component opens the possibility of regulating the system by modifying SecB levels or its interaction with HigA1 antitoxin. Do authors have any idea about possible mycobacterial peptides that could block Mtb-SecB-HigA1 interaction? Would it be possible to use the 3D structure of the Mtb-SecB/ChAD complex to look for such peptides? That would be an interesting finding and perhaps a new research avenue.

Reviewer #3 (Remarks to the Author):

The manuscript "Structural insights into chaperone addiction of toxin-antitoxin systems" by Guillet And colleagues, reveals some very interesting insights on the molecular mechanism of regulation of chaperone-dependent TAs. It also provides some missing clues as to how the antitoxins could be released from the chaperones and this might entails in context of TA regulation. Overall it provides a significant contribution to the field and much needed mechanistic description of this peculiar type of TA modules. However in its current form the text is hard to read or followed at times and additional efforts to better present the results will highly appreciated.

As a general remark, in the introduction it is stated that classical type II TA systems in general are involved in bacterial stress response and persistence. Although this has certainly being shown for a few special TA such as the chaperone-dependent TAs and a few modules present in Salmonella that are activated upon the entrance of the bacterium to the host macrophage, the general notion that the function of chromosomal TAs is related to bacterial stress adaptation have been recently heavily contested and it is no longer accepted. Therefore the statement in the introduction should revised an re-written in a more appropriate way. The methods section is also rather unbalanced. While the part on protein purification and crystallisation is very well described, the protocol used for the molecular dynamic simulation is completely missing (even the result of the simulation is not shown) and other parts such as the SAXS analysis and modelling is rather poor. Here the authors must provide a standard table with all the parameters obtained from SAXS besides R_g , D_{max} and X^2 , and the guide lines from Rambo and Tainer Nat. 2013 must be followed.

Another issue I have with the text in general, is the obsessive description of the details of the molecular interactions observed in the structures. From a mechanistic perspective, it is clear in some cases a detailed description of an interaction is required, however there are entire sections of the main manuscript that could be explained in a figure or are actually very well described in the figures of the paper (which I must say are very good). So I believe that the general reader of Nat Comm will appreciate if some of these parts are trimmed down.

Major comments:

- The authors use the term, active antitoxin, what do the mean by that?
- In some parts the authors used terms such as alpha-faces or alpha-interfaces that are not really corresponding to accepted structural features. Please revise that text throughout and correct this.
- The lack or presence of electron density is described in several part of the text. In this regard the authors should provide an omit electron density map contoured around the region of the peptide. This is common practice in structural biology.
- I find very interesting that the binding of the peptides to the chaperone is independent, specially since a transition from a dynamic to a more rigid state is involved. In this sense it would be

interesting to do a reversed titration and measure the n value and K_d to validate that the interaction sites are indeed independent.

- Also regarding the thermodynamic measurements, I wonder why only two sites are detected when it seems apparent that at least 4 sites are easily detected by mass spectrometry, could the authors design a titration to measure the binding of 4 peptides?
- The binding of the chaperone-antitoxin complex to DNA was also monitored by ITC, in this context it would be very very interesting if the authors could also measure the binding of the free antitoxin (or antitoxin DNA-binding domain) and the toxin-antitoxin to the same palindrome, in order to understand a bit better the interplay between all the players in transcription regulation.
- The dynamic state of the chaperone in the absence of the peptide as well as the folding state of HigA should be tested, given the relevance that the authors suggest this have. The dynamic state of the chaperone can not simply be assessed with these Kratky plots given that the overall structure of bound and unbound chaperone is very similar. Experiments of H-D exchange could help with this and if this is a very established fact of these type of protein it should be better explained in the text.
- In this sense, the authors state in the discussion that this is the first report of a SecB chaperone binding to a folded client. As it is, the statement is not properly supported by experimental data and is a bit misleading (for instance in the discussion the authors also refer to the toxin as "highly unstable"). While it is clear that HigA has a folded dimerisation domain that binds DNA, the folded state of the region that binds the chaperone is not described. Given that it has been shown that many antitoxins contain protein regions that are significantly disordered, the authors must show that this not the case for this particular antitoxin or tone down this statement.
- The proposed mechanism of the antitoxin release from the chaperone is one of the most exciting parts of the paper, given the implications this has in the regulation of the system. However the interaction of ChAD with the chaperone is on the weak side. In this regard it would be interesting if the authors could also measure the affinity of the chaperone for the full antitoxin or a longer version of the C-terminus of the antitoxin that is still soluble, to better understand how the ChAD peptide can titrate out the antitoxin.

Reviewer #4 (Remarks to the Author):

General Comment

The article presents a detailed structural and biophysical characterisation of key parts of a *Mycobacterium tuberculosis* SecB like toxin-antitoxin chaperone (TAC) system. The work concerns the SecB like chaperone (Mtb-SecBTA), that is known to bind a short region of the antitoxin (Mtb-HigA1), the "chaperone addition" sequence (ChAD). ChADh, destabilises the antitoxin in the absence of chaperone binding, thus rendering toxicity "chaperone mediated".

The authors here present detailed biophysical characterisation of Mtb-SecBTA both alone, and in complex with both ChAD, and also the full length antitoxin (Mtb-HigA1). New crystal structures of the Mtb-SecBTA tetramer and Mtb-SecBTA-3·ChAD are presented and compared in detail with previous structures of SecB homologs (Ec-SecB and Hi-SecB). The binding mode between ChAD and Mtb-SecBTA is compared and contrasted with published structures of SecB homologs bound to substrates. Detailed structural analysis and extensive mutagenesis is performed to identify the key binding residues and these mutants are tested *in vivo* for their toxin suppression ability in growth recovery assays.

Direct evidence of complex formation between Mtb-SecBTA and Mtb-HigA1 is presented using other biophysical methods, notably native MS, since this complex seems recalcitrant to crystallisation. Mtb-HigA1 is found to bind to Mtb-SecBTA as a dimer and this complex is found to bind a region of DNA *in vitro* corresponding to own repressor (repressor DNA binding is a known biological function). The authors use this as supporting evidence that the purified complex is in a functional state. Interestingly ChAD peptide is found to disrupt the interaction between HigA1 and

Mtb-SecBTA.

The TAC system under study here and the results presented are fundamentally important in understanding the regulation of HigA1 in this human pathogen, and undoubtedly interesting for a specialist reader, however the reviewer is unsure of the more general significance and wider relevance. Results are presented clearly and with a high level of technical detail, however it is often unclear to a non-specialist what the purpose of the experiments are as you progress through the article other than presenting a detailed biophysical/structural characterisation of the system. This kind of approach may be immediately tractable for a specialist audience, and suitable for a specialist journal, however, the reviewer suggests some reformatting and reorganising of the entire article, as well as a greater emphasis of putting the results into a wider biological context, would be necessary before publication in general interest journal such as Nature Comm (see below for detailed comment).

Reformatting for a more general readership

Each section would strongly benefit from a more gradual introduction describing what the authors are investigating, and many sections would benefit from some concluding sentences or interpretation describing the relevance of data presented.

Instances of this occur throughout the article, for example...

>The Results section jumps straight into a detailed interpretation of the SecB structure. Given that the Introduction section that immediately precedes it culminates with a summary of the findings of the article, a few introductory sentences here would be extremely useful. Why are we suddenly concerned with the structure of SecB, what are the authors hoping to discover here?

>Section Mtb-SecBTA/ChAD interaction. There is no introductory sentence here. What are the authors hoping to achieve by examining this interaction? A brief interpretation/summary at the end of the section would also be welcome.

>The section "Comparison of Mtb-SecB structures" contains a very detailed comparison of the respective crystal structures, but very little interpretation as to the biological or functional significance of the differences described. This would be extremely helpful, especially for a general readership.

> Section "Specific affinity of ChAD for Mtb-SecBTA chaperone". The author assumes that given the high-resolution crystal structures presented in previous sections, the purpose of this section is to characterise the interactions "in-solution" and build upon the static "snapshot" data already presented? Perhaps an introductory sentence to guide the reader would help here?

>Final paragraph – "We next used Native MS to investigate whether a stably formed... complex is could be disrupted by increasing concentrations of ChAD peptide." What is the biological relevance of this experiment, what are the authors trying to demonstrate by it and on what basis?

>Discussion, the comparison of the binding mode between Mtb-SecB and other SecB homologs is made. How significant/surprising are the differences observed the given the different biological functions and specificities of the chaperones

Detailed comments

Introduction

Page 1 – "(in about 7% of the cases)" change to "(in about 7% of cases)"

Page 1 – "It is believed that under certain stress the less stable..." suggest change to "It is believed that under certain stress conditions the less stable..."

Page 1 – “...thus inhibiting growth until normal growth condition resume.” Change to “...thus inhibiting growth until normal growth conditions resume.” Or reword

Page 2 – sentence on “drug persists” – this is an important point to demonstrate the wider relevance of the work presented. The article may benefit from describing the role of TAC in drug resistance in more detail here.

Page 2 – “...RelE toxin superfamily of ribonuclease.” Change for “...RelE toxin superfamily of ribonucleases.”

Results

Section on MtB-SecBTA/ChAD interaction. Only 3 ChAD are found bound in this highly symmetrical structure rather than the expected 4, could the authors speculate as to why this occurs?

Similar question for Calcium, that is found in 3 but not 4 subunits?

Section “Specific affinity of ChAD for MtB-SecBTA chaperone”. The reviewer agrees with the author that the binding of higher numbers of ChAD is likely non-specific, especially due to the high concentrations of ChAD used in binding experiments here (0-195uM from supp info). It should be noted somewhere in the article/supp, that concentrations of substrate+enzyme > ~50uM likely lead to non-specific adduction in the electrospray process. Ref – “Protein Complexes in the Gas Phase: Technology for Structural Genomics and Proteomics” Chem. Rev., 2007, 107 (8), pp 3544–3567 and others.

Correction for this has been proposed when calculating KD values “A method for removing effects of nonspecific binding on the distribution of binding stoichiometries: application to mass spectroscopy data.” Biophys J. 2010 Sep 8; 99(5):1645-9. The authors wondered if this correction could be applied to test for, and quantify, the non-specific adduction in their data? And even perhaps calculate more accurate kD values?

The results state that KD's of 9-23uM could be calculated. Could the authors please confirm if these are microstate KDs and if so add the used in Unidec to calculate them including the model used and justification, to the supp info.

For the native MS data, the deconvoluted spectra show binding to the tetramer and hexameric complex in Figs 4 S4 and S5. The spectra in these figures do not cover the mass range required to also see the monomer/lower order oligomers that may also be present. Could the authors confirm if any of these lower order oligomers were observed and if so how binding of ChAR to them was accounted for in the data analysis? The reviewer strongly suggests that the raw data be made available for examination by the reader in any final publication.

Brief question, did the native MS results indicate the presence of calcium as suggested by the crystal structure? (the reviewer acknowledges that this may not have been resolvable)

Section “Full length MtB-SecTA/HigA1 chaperone/antitoxin complex”. Authors use SEC-MALS to confirm interaction of MtB-SecTA/HigA1 hexamer with DNA, which likely forms a 4:2:1 complex and also confirm by ITC which indicates an association of 1.1. Could the authors confirm the error in the later(1.1)? Was it possible to verify this result using nativeMS which would provide much more certainty to the complex stoichiometry and presumably indicate if indeed there is “binding of slight more than one palindromic DNA sequece per MtB-HigA1 dimer.”

Final paragraph. The authors interpret the dissociation of the MtB-SecTA/HigA1 hexamer upon addition of ChAD, to binding to the hexamer, followed by dissociation. Given that much binding in the nativeMS experiments may be non-specific, could the authors please justify their interpretation here? Another explanation, that may be equally likely given the data presented, is that since the MtB-SecTA/HigA1 exists in equilibrium with free- MtB-SecTA, addition of ChAD may also bind directly to MtB-SecTA. If this MtB-SecTA.ChAD cannot bind HigA1 this would likely shift the MtB-

SecTA/HigA1 vs free-MtB-SecTA equilibrium towards the tetramer thus favouring hexamer dissociation. Could the authors comment in the text on whether this mechanism may also occur?

Native MS Methods

The cone voltage of 150V seems very high for native measurements, and especially for binding measurements could the authors please comment?

The cap voltage is said to be 1.6kV is this in addition to the potential applied directly to the nanomate chip?

Subtract curve in UniDec is known to potentially greatly distort intensities of CS distributions in distinct m/z ranges, present in the same spectrum (such as the tetramer/ hexamer here). The value for this parameter should remain quite constant throughout the same data set for fair interpretation and comparison. Have the authors examined the effect of this parameter on relative intensities and perhaps performed manual validation on at least one data point to check appropriateness?

Using a FWHM peak width of 1Th will likely hugely overfit the data (peaks on Q-ToF more likely 10-100Th wide) resulting in intensity abnormalities, the reviewer suggests using a more realistic peak width for data processing.

REVIEWER #1 (REMARKS TO THE AUTHOR):

The manuscript of Guillet and colleagues describes some structural insights into the chaperone addiction of the *M. tuberculosis* HigBA. In the first part, the authors provide a thorough structural study of the Mtb SecB tetramer in complex with the ChAD peptide of the HigA1 antitoxin. The second part describes their various biochemical attempts at deciphering the architecture of the Mtb SecB/HigA1 complex. Whereas the structural part is conclusive, the biochemical part is rather ambiguous and might require further reconsideration. Thus, I have major concerns on the quality of the manuscript.

Major concerns:

Each of the different experiments (structural investigations, ITC, native MS using the ChAD peptides, and SEC-MALS) give different stoichiometry of HigA1 or ChAD peptide bound to tetrameric SecB.

1) The authors argue that in the crystal structure steric clashes prevent a fourth peptide molecule binding to SecB. It remains unclear whether this is simply due to crystal contacts or to the distorted D₂ symmetry. Along this line, it's also unclear why the authors do not see a Calcium ion bound to chain C, although a peptide is bound. To which extent are these results biased by crystal artifacts?

This is an important point raised by the four reviewers. As described in our manuscript and detailed in Supplementary Table 2, each of the three ChAD peptides found in the structure interacts with three subunits of the *Mtb*-SecB^{TA} tetramer, which correspond to three interfaces. Two of these three interfaces mostly contribute to peptide binding. They involve residues D17, R29, and L108 from one subunit and residues Y49, I77, L154, P155, P156 and L157 from another subunit. Crystal packing analysis revealed that these residues are accessible in all subunits, including chains B and D that should interact with the fourth peptide. Thus, steric clashes due to the distorted D₂ symmetry prevent a fourth peptide molecule from binding to SecB. This has been clarified in our revised version and as requested by both reviewers 1 and 3, a new figure showing unbiased electron density map for all three peptides and in the vicinity of what would be a fourth binding site has been added to the manuscript as Supplementary Fig. 4.

We also checked packing constraints around the calcium-binding site, which includes S65, and found that this site is fully accessible in subunits A and B whereas the loop β 4- α 1 of symmetry mates is found in its vicinity for subunits C and D. In the case of subunit D, the side chain of residue D118 from the symmetry mate interacts with and stabilizes S65 but does not prevent calcium binding. In contrast, the side chain of residue E115 from another symmetry mate blocks the calcium-binding site in subunit C. A sentence has also been added in the new version of the manuscript to clarify this point.

See also our responses to reviewers 2 and 4.

Text modifications:

Please note that page numbering corresponds to the marked up version of the revised manuscript

- Page 7: "The lack of calcium binding to subunit C is due to crystal packing constraint where loop β 4- α 1 of a symmetry mate partly occupies the site."

- Pp. 9-10: “On the other hand, no sufficient interpretable electron density could allow the building of a symmetrical equivalent of peptide G (Supplementary Fig. 4), which because of steric clashes due to the distorted D_2 symmetry would not be possible without structural rearrangement.”

2) ITC experiment Figure 4a. The design of the experiment in its current form does not allow to unambiguously determine the molar ratio of ChAD peptide and SecB. One might consider to vary the ratio ChAD to SecB in such experiments.

We initially performed several additional ITC experiments using different ChAD/SecB ratios. All of them unambiguously converged toward a stoichiometry consistent with the binding of two peptides per *Mtb*-SecB^{TA} tetramer. We have replaced Fig. 4a with a new panel where ITC was performed with 38 μ M SecB and 204 μ M ChAD peptide, which gives an improved profile.

3) Furthermore, if the ITC experiment is correct (2 binding sites), why do the authors observe 3 peptide molecules bound in the crystal structure?

See our answer below.

4) Why do the authors observe 4 ChAD peptides binding to SecB in native MS experiments? This is again in strong contrast to all previous results.

In fact, we can only hypothesize that these differences are linked to the different ChAD/SecB ratios used in our experiments. For instance, an excess of 3 peptides per *Mtb*-SecB^{TA} monomer was used for crystallization while we went up to a ratio of 20 peptides per protein monomer for native MS. That high molar ratios of ChAD peptide could lead to non-specific binding in native MS is now discussed in the text as suggested by reviewer 4. However, this might only concern the low-abundance interactions observed for molar ratios >4 . Indeed, the four relatively high-affinity events, occurring at ratios <4 , are rather consistent with the crystallographic observations, which revealed the binding of three peptides and a potential fourth binding site for which only weak electron density could be observed (as mentioned in the original version of our manuscript and now illustrated in the Supplementary Figure 3 of the revised version). As stated above, all ITC experiments identified two binding sites, whatever the conditions used. However, the peptide/protein monomer ratio achieved at the end of the titration used for instance to illustrate Fig. 4a (new experiment as recommended by reviewer 1) was 1.0. Using higher ratios (in our case up to 2.5) lead to shifting the curve, which - as noticed by the reviewer - does not allow the unambiguous determination of stoichiometry. Finally, the possible implication of four binding sites with respect to the binding of *Mtb*-HigA1, where each protomer of the antitoxin dimer could cover two binding sites of the chaperone, has been discussed in the last paragraph of the Discussion.

5) Eventually, once the authors use full-length HigA1 they observe a 4:2 ratio of SecB to HigA1. Which of the 3 (4?) binding sites are occupied?

Indeed, the only molecular species observed from SEC/MALS or native MS experiments on purified *Mtb*-SecB^{TA}/HigA1 complex correspond to a 4:2 ratio of SecB to HigA1. However, as mentioned above, we do not exclude that all four binding sites could be occupied. The only way to prove it would be to solve the *Mtb*-SecB^{TA}/HigA1 structure. Unfortunately, the complex has been reluctant to crystallization so far.

The molecular biology experiments on mutated SecB expressed in *E. coli* are difficult to understand. It remains unclear to the reviewer to which extent

the TAC part, the chaperone activity of SecB, or the expression rate and protein stability are causing the observed differences.

As requested, we have monitored the steady state expression of all the mutants used in this work. A supplementary figure has been included in the revised version (Supplementary Fig. 6).

Text modifications:

- Page 10: “Steady state expression of the *Mtb*-SecB^{TA} derivatives revealed that all mutants are well expressed when compared to the wild-type (Supplementary Fig. 6).”

- Page 34: “Supplementary Figure 6: Expression of the *Mtb*-SecB^{TA} mutants. Transformants of strain W3110 Δ *secB* containing p29SEN-*Mtb*-SecB^{TA} mutant derivatives were grown at 37 °C in LB Ampicillin. At mid-log phase, IPTG inducer (500 μ M) was added 2 h before preparing whole cell extracts. Extracts were separated on mini-PROTEAN 4-15% TGX precast gels (Bio-Rad) and steady state protein expression was visualized following western blotting using anti-*Mtb*-SecB^{TA} antibodies as previously described (Bordes et al., 2011). Arrows indicate the presence of *Mtb*-SecB^{TA} derivatives.”

Minor comments:

The authors should show somewhere in the manuscript unbiased electron density maps for all 3 peptides.

As requested, we have prepared a new figure showing unbiased electron density map for all three peptides and in the vicinity of what would be a fourth binding site (Supplementary Fig. 4).

Text modifications:

- Page 9: “For all three peptides, there was no adequate electron density to build the last residue (G116) and the side chain of R115 is also missing in peptide G (Supplementary Fig. 4).”

- Pp. 9-10: “On the other hand, no sufficient interpretable electron density could allow building of the symmetrical equivalent of peptide G (Supplementary Fig. 4), which because of steric clashes due to the distorted D₂ symmetry would not be possible without structural rearrangement.”

- Page 33: “Supplementary Figure 4: Electron density map around the ChAD peptide. The feature enhanced map, calculated after having deleted atoms corresponding to the peptides, contoured at 1.0 sigma is shown in cyan. (a) Chain E. (b) Chain F. (c) Chain G. (d) Around what would be a fourth peptide-binding site where a 2-fold symmetry mate of peptide G has been generated through the DCBA permutation.”

In some aspects it would be helpful to the reader if figures would directly indicate which protein is expressed by which inducer.

We agree with the reviewer and we have added illustrations depicting the promoters and the genes encoding the proteins that are expressed in Fig. 3, Supplementary Fig. 5 and new Supplementary Fig. 6.

The comparison of Mtb SecB with related pdb entries is lengthy and the authors might consider shortening this part.

As requested by all reviewers, this part and others have been substantially shortened or revised.

REVIEWER #2 (REMARKS TO THE AUTHOR):

In their study Guillet and colleagues resolved the structure of the Mycobacterium tuberculosis SecB homolog bound to the HigA1 antitoxin C-terminal peptide ChAD. Type II toxin-antitoxin (TA) systems are bicistronic operons encoding a toxic protein that block crucial processes in the bacterial cell (i.e. translation, DNA replication), and an antitoxin that neutralizes its poisonous partner. Mycobacterial HigBA1 is a special type II TA where there exists a third component, the chaperone SecB, that assists the folding of the antitoxin HigA1 and allows the proper neutralization of the toxin HigB1. Yet this process was previously described, the structural details of interaction between SecB and HigA1 were unknown. In this study the authors resolved the crystal structure of SecB-ChAD to further understand how SecB chaperones can discern between substrates. The structure of SecB-ChAD shows major differences on binding compared to other SecB substrates (e.g. SecB-SecA). Guillet and colleagues identified key residues of SecB necessary to bind the antitoxin C-terminal peptide ChAD. It is also suggested that HigA1 antitoxin can reach an active form while bound to SecB. A large variety of techniques are used to validate these findings.

I personally think that the work presented by Guillet and colleagues has quality enough to be published in this journal. Authors use a large variety of techniques to validate the crystal structure of Mtb-SecB/ChAD. Also, the manuscript is clearly written and the data well presented. I only have a few questions and some comments that I think might clarify some parts of the text / Figures; I list them below:

1- "In absence of stress, the antitoxin-bound toxin is inactive and the bacteria grow normally." Although it is true that TA systems have been associated to different stress conditions, including antibiotic exposure, heat shock or pH, TA systems are also widely distributed in plasmids, where they mediate plasmid stability. In this case there is no stress per se, but it is the lost of the plasmid the event that impairs the synthesis of new antitoxin and therefore the activation of the toxin. I think this concept has to be introduced at this point of the introduction for readers not familiarized with the topic. Otherwise a biased impression of the TA systems is transmitted.

We agree with the reviewer that plasmid stability mediated by TA systems is a key aspect of TA physiology. As requested, sentences have been added in the introduction.

Text modifications:

- Pp. 3-4: "TA systems are also widely distributed on plasmids where they contribute to their stability. In this case, when the TA-containing plasmid is lost, *de novo* synthesis of the antitoxin stops and the reservoir of cytosolic antitoxin is rapidly degraded, which results in toxin activation and cell growth inhibition (Yamaguchi et al., 2011)."

Yamaguchi, Y., Park, J.H., and Inouye, M. (2011). Toxin-antitoxin systems in bacteria and archaea. Annu Rev Genet 45:61-79.

2- "Accordingly, such a reversible control of bacterial growth by toxins has been involved in bacterial adaptation to stress, persistence and virulence^{13,14}." Change reference 14 by Lobato-Márquez et al., 2016, FEMS Microbiol Rev, where a comprehensive revision of TA modules and its connection to virulence is provided.

Reference 14 has been changed as suggested by the reviewer.

3- "We also noted that the presence of calcium ions was important for the crystallization of the complex." What is the relevance of Ca²⁺ ions on SecB structure? What is the role of DMSO molecules on protein stabilization? Are they also important in SecB structure alone?

As mentioned in the submitted manuscript, Ca²⁺ ions may play a structural role in the case of *Mtb*-SecB^{TA} through stabilizing the structure and/or the crystal packing but we have no evidence that they could play a functional role. In fact, *Ec*-SecB has also been crystallized in the presence of CaCl₂ (80 mM) but no Ca²⁺-binding site was described in the corresponding article (Dekker et al., 2003). In addition, our native MS analysis did not reveal any Ca²⁺ adduct when *Mtb*-SecB^{TA} or *Mtb*-SecB^{TA}/ChAD was desalted in the presence of high concentration of calcium acetate, i.e. up to 360 mM. See also our responses to reviewers 1 and 4.

As mentioned in our manuscript, DMSO was previously shown to be necessary for the growth of diffraction-quality crystals of *Mtb*-SecB^{TA} but was not sufficient to help resolving the structure (Lu et al., 2016). We found that the use of DMSO was instrumental in improving initial needle-like crystals and allowed collecting diffraction data to 3.1 Å resolution, which however did not allow solving the structure. The combined use of DMSO and calcium acetate, and perhaps more importantly, the fact that *Mtb*-SecB^{TA} was crystallized in the presence of the ChAD peptide were important factors for the successful crystal structure determination at ca. 1.8 Å resolution. Furthermore, we showed by differential scanning fluorimetry (DSF) that DMSO slightly destabilizes *Mtb*-SecB^{TA} (ΔT_m values of -1.2 and -4.2 °C in the presence of 5 and 20% DMSO, respectively), in accordance with the described mechanism where DMSO binding must be enhanced upon protein destabilization/unfolding (Arakawa et al., 2007). These DSF experiments have not been included in the submitted version because of their limited interest.

Arakawa, T., Kita, Y., and Timasheff, S.N. (2007). Protein precipitation and denaturation by dimethyl sulfoxide. Biophys Chem 131:62-70

Dekker, C., de Kruijff, B., and Gros, P. (2003). Crystal structure of SecB from Escherichia coli. J Struct Biol 144:313-9

Lu, Z., Wang, H., and Yu, T. (2016). The SecB-like chaperone Rv1957 from Mycobacterium tuberculosis: crystallization and X-ray crystallographic analysis. Acta Crystallogr F Struct Biol Commun 72:457-61

4- In the Results paragraph where the differences between the *Mtb*-SecB^{TA} tertiary structure and its SecB counterparts are described, do the authors know how important are these regions for *Mtb*-SecB^{TA} function? Have authors tried to create chimeras of *E. coli* SecB and *Mtb*-SecB^{TA} for example, and test them in toxicity-neutralization or SecB chaperone generic ability assays?

Although we agree that such experiments would be interesting, the structure of SecB is compact and insertion or mutations in the β -sheet or in the long helix $\alpha 1$ generally lead to SecB aggregation or instability both *in vitro* and *in vivo*, which makes such chimeras very hard to rigorously investigate. In addition, the fact that *Ec*-SecB and *Mtb*-SecB^{TA} only show 15% sequence identity renders such approaches even more risky. Since previous work showed that single mutations in *Ec*-SecB are sufficient to redirect the chaperone to the TA systems without affecting export or interaction with SecA (Sala et al., 2017), it is reasonable to assume that such structural features may not be essential for *Ec*-SecB-mediated TA control and for export.

Sala, A.J., Bordes, P., Ayala, S., Slama, N., Tranier, S., Coddeville, M., Cirinesi, A.M., Castanie-Cornet, M.P., Mourey, L., and Genevoux, P. (2017). Directed evolution of SecB chaperones toward toxin-antitoxin systems. *Proc Natl Acad Sci U S A* 114:12584-9

5- In the Results *Mtb*-SecB^{TA} / ChAD interaction section, do the authors know if the binding of peptides E, F and G occur at same time? Could it be a cooperative binding where interaction of peptide E increases the affinity for peptide F and binding of F for G?

The native MS analysis performed with Unidec suggests that the binding sites are independent, i.e. there is no cooperative binding.

6- "All the *Mtb*-SecB^{TA} mutations that most significantly impaired TA control, with the exception of Y49A, additionally affected its ability to..." I do not understand why authors focus so much in the mutant Y49A when other mutants, such as D27A or F67A have similar neutralization ability in experiments shown in Fig.3.

We apologize for the mix-up. We focused on Y49A since, unlike other mutations (including D27A and F67A), the replacement of the tyrosine at position 49 did not affect the ability to replace the *E. coli* export chaperone SecB *in vivo* in the conditions tested. The corresponding sentence has been slightly modified.

Text modifications:

- **Page 11: "All the *Mtb*-SecB^{TA} mutations that most significantly impaired TA control additionally affected the ability to replace the *E. coli* export chaperone SecB *in vivo* at low temperature of growth, with the exception of Y49A (Supplementary Fig. 5)."**

7- How authors explain the gain of function of SecB mutants in chaperone generic ability? Is it due to a change of substrate specificity?

Based on our previous experience with gain/loss of generic function *in vivo*, we believe that subtle changes on SecB surface could indeed increase export function, either by modulating substrate specificity on by improving interaction with its SecA partner. A comment has been added in the revised version.

Text modifications:

- **Page 11: "Whether these mutations affect SecB substrate-binding specificity or its interaction with its SecA partner is not known."**

8- Please indicate in Results section what ITC and MST stand for the first time these two techniques are mentioned in the text.

ITC and MST have been defined at their first occurrence in the revised text (Results section).

9- When authors analyzed the quaternary complexes between and Mtb-SecB^{TA} and HigA1 by SEC-MALS and Mass-spec they found that SecB tetramer interacts with a HigA1 dimer. I might have missed something but, how is this result compatible with the crystal structure shown in Fig.1A where SecB was found to interact with 3 ChAD peptides? Is this an artifact of the crystals and SecB can only bind 2 ChAD peptides and therefore 2 antitoxin molecules? Have authors tried different buffer concentration (e.g. increasing ionic strength) to break possible unspecific interactions? I do not obviously find this issue discussed anywhere in the text.

We discuss the important point of stoichiometry raised by the four reviewers in several instances of our revisions. Concerning the specific question of reviewer 2 related to the impact of ionic strength on *Mtb*-SecB^{TA}/ChAD interaction, this was impossible to investigate using native MS since NaCl is not volatile and therefore non-MS compatible. In addition, SEC-MALS analyses did not lead to unambiguous conclusion, despite numerous samples of complex injected, and because the variation in molar mass is certainly too small (note: MW of ChAD=1,588 Da, MW of the *Mtb*-SecB^{TA} tetramer=81,552 Da).

10- In TA systems the binding of the antitoxin to the DNA promoter operator regions is modulated by the toxin:antitoxin ratios. How does the toxin HigB1 influence the DNA-binding capacity of the complex SecB-HigA1? How strong is the binding of HigA1 alone (in the absence of Mtb-SecB) to DNA?

The robust aggregation-prone property of *Mtb*-HigA1 renders the second proposed experiment very difficult to tackle. Binding of the antitoxin alone to its promoter region has been tested by another group (Fivian-Hughes & Davis, 2010). As shown in own work (Bordes et al., 2011), the authors found that *Mtb*-HigA1 was highly insoluble in the absence of the chaperone and impossible to purify in a soluble form unless extracted from the aggregated fraction with urea. In this case, the authors could still show that *Mtb*-HigA1 indeed specifically binds the *higB* P2 promoter region, although in this case very high amounts of proteins (up to 10 μ M) were necessary to detect binding. Thus, these data somehow suggest that refolded *Mtb*-HigA1 alone might have a low affinity for DNA but we find it hard to draw rigorous conclusions. Please note that reviewers 3 and 4 have also raised this point.

Fivian-Hughes, A.S., and Davis, E.O. (2010). Analyzing the regulatory role of the HigA antitoxin within *Mycobacterium tuberculosis*. *J Bacteriol* 192:4348-56

Bordes, P., Cirinesi, A.M., Ummels, R., Sala, A., Sakr, S., Bitter, W., and Genevoux, P. (2011). SecB-like chaperone controls a toxin-antitoxin stress-responsive system in *Mycobacterium tuberculosis*. *Proc Natl Acad Sci U S A* 108:8438-43

Concerning DNA binding, we have now confirmed by native MS experiments the formation of a chaperone/antitoxin/DNA heptameric 4:2:1 complex when mixing *Mtb*-SecB^{TA}/HigA1 and 38-bp palindromic sequence derived from the *higB* P2 promoter (see Supplementary Fig. 9a), as requested by reviewer 4. We agree with reviewer 2 that the impact of the toxin on the antitoxin or on the antitoxin-chaperone complex, with or without DNA, is a very interesting aspect of TAC systems. Yet, this has been very difficult to address so far due, in part, to the difficulty to express the toxin to high level and high

purity grade, perhaps because of its high toxicity. Nevertheless, the structure and the dynamics of these complexes with or without DNA is something that we prefer to address independently on a long-term basis, as we believe that it deserves a thorough investigation both *in vitro* and *in vivo*.

11- In the Material and Methods section "In vivo TA control by chaperones" authors should indicate which inducers induce which gene. Fig.3 can be confusing.

The Methods section entitled “*In vivo* TA control by chaperones” has been revised accordingly on pp. 18-19.

12- Figure 3a: How many times were these spot assays done? This is not mentioned anywhere.

Please indicate what genes are induced with each inducer in the Figure legend in the same way that is mentioned in Figure 5d.

Also, I would suggest adding lines close to the inducer concentrations indicating which set of strains were spotted under each inducer (Same as Fig. 5d).

Spot tests have been performed at least in triplicates. This is now mentioned in the legend of the figure, which has also been modified to indicate what genes are induced with each inducer (as in Fig. 5d). Indication of the gene expressed has also been added on all *in vivo* figures as requested by both reviewers 1 and 2.

13- Figure 3b: I think changing the color of SecB to a lighter grey would allow distinguishing the residue numbers more easily.

The color of *Mtb*-SecB^{TA} has been changed to a lighter grey as recommended. To be distinguishable, residues that were not tested (i.e. Ala residues and Y146) are now colored in pale yellow, instead of white in the previous version. The legend of the figure has been changed accordingly.

14- Figure 6: To be consistent adapt the color of the SecB interacting protein in the Hi-SecB/SecAc upper and bottom panels as done for Ec-SecB/PhoA and Mtb-SecB/ChAD.

For clarity reason, SecAc is now shown on both sides of SecB in upper panel, as found in the structure of the SecB/SecAc complex, and we kept the preprotein in green as in panel (a) because it is more reflective of the *in vivo* situation.

15- Supplementary Figure 2: to make this figure cleared authors should consider putting the names of the Sec complexes in the Figure.

The names of the Sec complexes have been introduced in the Figure.

16- Supplementary Figure 3: It would be helpful if authors explained the basis for the novobiocin assay either on the Results or Material and Methods section. Although this is referred to other papers, it is not explain there either.

As requested, we have now explained the basis for the novobiocin assay in the Methods section “*In vivo* chaperone assays”.

Text modifications:

- Page 19: “It was previously shown that *E. coli secB* mutants do not grow at low temperature and are sensitive to novobiocin antibiotic (Nichols et al., 2011; Sala et al., 2017; Ullers et al., 2007), likely due to the intrinsic cold sensitive property of the Sec translocon and to an impaired outer membrane (Pogliano & Beckwith, 1993; Sala et al., 2014). In this case, it was found that growth at low temperature and resistance to novobiocin requires a functional interaction between SecB and SecA *in vivo* (Sala et al., 2017; Ullers et al., 2007).”

Nichols, R.J., Sen, S., Choo, Y.J., Beltrao, P., Zietek, M., Chaba, R., Lee, S., Kazmierczak, K.M., Lee, K.J., Wong, A., Shales, M., Lovett, S., Winkler, M.E., Krogan, N.J., Typas, A., and Gross, C.A. (2011). Phenotypic landscape of a bacterial cell. *Cell* 144:143-56

Pogliano, K.J., and Beckwith, J. (1993). The Cs sec mutants of *Escherichia coli* reflect the cold sensitivity of protein export itself. *Genetics* 133:763-73

Sala, A., Bordes, P., and Genevax, P. (2014). Multitasking SecB chaperones in bacteria. *Front Microbiol* 5:666

Sala, A.J., Bordes, P., Ayala, S., Slama, N., Tranier, S., Coddeville, M., Cirinesi, A.M., Castanie-Cornet, M.P., Mourey, L., and Genevax, P. (2017). Directed evolution of SecB chaperones toward toxin-antitoxin systems. *Proc Natl Acad Sci U S A* 114:12584-9

Ullers, R.S., Ang, D., Schwager, F., Georgopoulos, C., and Genevax, P. (2007). Trigger Factor can antagonize both SecB and DnaK/DnaJ chaperone functions in *Escherichia coli*. *Proc Natl Acad Sci U S A* 104:3101-6

17- Supplementary Figure 5: authors should consider changing the order of the graphs so "a" is followed by "b" and "c" horizontally at the top and "d" and "e" horizontally at the bottom.

We have prepared a new figure (Supplementary Fig. 9 in the revised version) with a new graph "a" corresponding to the MS analysis of *Mtb*-SecB^{TA}/HigA1 in the presence of DNA and where graphs "b" to "f" (formerly "a" to "e") can now be read horizontally. Although not requested by the reviewer, we have also modified Supplementary Fig. 8 (Supplementary Fig. 4 in the submitted version) the same way.

18- Have the authors tested the specificity of Ec-SecB or Hi-SecB to ChAD?

This is an interesting point that we have recently addressed in a directed evolution experiment of *Ec*-SecB toward TA systems where we found, both *in vivo* and *in vitro*, that only the evolved SecB mutants capable of efficiently inhibiting the toxin *in vivo* were able to specifically bind to the ChAD region of *Mtb*-SecB^{TA} (Sala et al., 2017).

19- Following question 18, authors describe mutations that affect *Mtb*-SecB binding to HigA1 but increase the general chaperone function of *Mtb*-SecB. Have authors tried adapting those equivalent residues of Ec-SecB or Hi-SecB and changing them into the sequence of *Mtb*-SecB? By doing that they might see an increase in Ec-SecB or Hi-SecB binding to ChAD. This kind of experiments might elucidate extra information about SecB chaperones specificity for their partners.

This specific point was addressed by Sala et al. (2017) by randomly mutating *Ec*-SecB and by subsequently selecting SecB mutants that are specifically directed toward *Mtb*-

HigBA1. Remarkably, in this case single mutations in SecB were sufficient to increase its ability to control *Mtb*-HigBA1. Such mutations largely map in the major substrate binding area of SecB and are presented in Supplementary Figure 10 (former Supplementary Figure 6) in the legend of which the reference (Sala et al., 2017) has been added.

Sala, A.J., Bordes, P., Ayala, S., Slama, N., Tranier, S., Coddeville, M., Cirinesi, A.M., Castanie-Cornet, M.P., Mourey, L., and Genevoux, P. (2017). Directed evolution of SecB chaperones toward toxin-antitoxin systems. *Proc Natl Acad Sci U S A* 114:12584-9

20- In most type II TA systems, the antitoxin C-terminal domain is normally unstructured, and only gets structured when the antitoxin binds the toxin. I was wondering if in TAC systems the chaperone binding domains are somehow connected to the "classic" unstructured C-terminal domain. Could it be an evolution of TA systems? Authors might comment this in the Discussion section.

We agree with the reviewer. However, with too little data available at this stage of our work, we would prefer not to be too speculative about this specific point. In addition, although we think it might be the case, we do not have any evidence that the C-terminal ChAD is unstructured (using available bioinformatics tools), although it confers aggregation properties to the antitoxin.

21- Until recently it was believed that the main regulation of antitoxin levels in type II TAs occurred at the transcriptional level and at the protein level via protease (e.g. Lon, Clp) degradation. Here, the presence of a third component opens the possibility of regulating the system by modifying SecB levels or its interaction with HigA1 antitoxin. Do authors have any idea about possible mycobacterial peptides that could block *Mtb*-SecB-HigA1 interaction? Would it be possible to use the 3D structure of the *Mtb*-SecB/ChAD complex to look for such peptides? That would be an interesting finding and perhaps a new research avenue.

We fully agree with the reviewer and we have now clearly expressed this possible aspect of our work, as also suggested by reviewer 4 (see comment below). Of course, one of the main perspectives will be to use the current structure to develop peptides that could disrupt such complex and trigger toxin release.

REVIEWER #3 (REMARKS TO THE AUTHOR):

The manuscript "Structural insights into chaperone addiction of toxin-antitoxin systems" by Guillet and colleagues, reveals some very interesting insights on the molecular mechanism of regulation of chaperone-dependent TAs. It also provides some missing clues as to how the antitoxins could be released from the chaperones and this might entails in context of TA regulation. Overall it provides a significant contribution to the field and much needed mechanistic description of this peculiar type of TA modules. However in its current form the text is hard to read or followed at times and additional efforts to better present the results will highly appreciated.

As a general remark, in the introduction it is stated that classical type II TA systems in general are involved in bacterial stress response and persistence. Although this has certainly being shown for a few special TA such as the chaperone-dependent TAs and a few modules present in *Salmonella* that are activated upon the entrance of the bacterium to the host macrophage, the general notion that the function of chromosomal TAs is related to bacterial stress adaptation have been recently heavily contested and it is no longer accepted. Therefore the statement in the introduction should revised an re-written in a more appropriate way.

We have revised the introduction to take into account comments from reviewers 1 and 3.

Text modifications:

- Pp. 3-4: “TA systems are also widely distributed on plasmids where they contribute to their stability. In this case, when the TA-containing plasmid is lost, *de novo* synthesis of the antitoxin stops and the reservoir of cytosolic antitoxin is rapidly degraded, which results in toxin activation and cell growth inhibition (Yamaguchi et al., 2011). Control of bacterial growth by toxins has been associated with various cellular processes, including stabilization of genomic regions, protection against foreign DNA, biofilm formation, control of the stress response, bacterial virulence and persistence (Lewis, 2010; Lobato-Marquez et al., 2016). Although an involvement of TA systems in persistence was found for several bacteria (Cheverton et al., 2016; Helaine et al., 2014; Norton & Mulvey, 2012; Van Acker et al., 2014) their contribution to *Escherichia coli* K-12 drug persisters has not been demonstrated (Goormaghtigh et al., 2018; Harms et al., 2017).”

Cheverton, A.M., Gollan, B., Przydacz, M., Wong, C.T., Mylona, A., Hare, S.A., and Helaine, S. (2016). A Salmonella Toxin Promotes Persister Formation through Acetylation of tRNA. *Mol Cell* 63:86-96

Goormaghtigh, F., Fraikin, N., Putrins, M., Hallaert, T., Hauryliuk, V., Garcia-Pino, A., Sjodin, A., Kasvandik, S., Udekwu, K., Tenson, T., Kaldalu, N., and Van Melderen, L. (2018). Reassessing the Role of Type II Toxin-Antitoxin Systems in Formation of *Escherichia coli* Type II Persister Cells. *MBio* 9

Harms, A., Fino, C., Sorensen, M.A., Semsey, S., and Gerdes, K. (2017). Prophages and Growth Dynamics Confound Experimental Results with Antibiotic-Tolerant Persister Cells. *MBio* 8

Helaine, S., Cheverton, A.M., Watson, K.G., Faure, L.M., Matthews, S.A., and Holden, D.W. (2014). Internalization of Salmonella by macrophages induces formation of nonreplicating persisters. *Science* 343:204-8

Lewis, K. (2010). Persister cells. *Annu Rev Microbiol* 64:357-72

Lobato-Marquez, D., Diaz-Orejas, R., and Garcia-Del Portillo, F. (2016). Toxin-antitoxins and bacterial virulence. *FEMS Microbiol Rev* 40:592-609

Norton, J.P., and Mulvey, M.A. (2012). Toxin-antitoxin systems are important for niche-specific colonization and stress resistance of uropathogenic *Escherichia coli*. *PLoS Pathog* 8:e1002954

Van Acker, H., Sass, A., Dhondt, I., Nelis, H.J., and Coenye, T. (2014). Involvement of toxin-antitoxin modules in *Burkholderia cenocepacia* biofilm persistence. *Pathog Dis* 71:326-35

Yamaguchi, Y., Park, J.H., and Inouye, M. (2011). Toxin-antitoxin systems in bacteria and archaea. *Annu Rev Genet* 45:61-79

The methods section is also rather unbalanced. While the part on protein purification and crystallisation is very well described, the protocol used

for the molecular dynamic simulation is completely missing (even the result of the simulation is not shown) and other parts such as the SAXS analysis and modelling is rather poor. Here the authors must provide a standard table with all the parameters obtained from SAXS besides R_g , D_{max} and X^2 , and the guide lines from Rambo and Tainer Nat. 2013 must be followed.

A paragraph has been introduced in the “Methods” section to describe the protocol used for the molecular dynamics calculations and a new supplementary figure (Supplementary Fig. 2) showing the result of the simulation has been prepared.

Concerning SAXS, we used, as suggested by the reviewer and in addition to Guinier analysis, the Rambo and Tainer concentration-independent method (Rambo & Tainer, 2013). Both methods gave consistent results. Furthermore, modeling was followed with classical χ^2 parameter, as inspection of the errors showed that the mean errors in the last third of all three scattering curves were in the order of 10%, where χ^2 and χ^2_{free} should behave similarly.

Text modifications:

- Page 22: “Molecular dynamics calculations. The X-ray structure was used to generate a starting template of the *Mtb*-SecB^{TA}/ChAD complex after building missing residues of the chaperone (Table 1), finally accounting for 161 residues from Asp6 to Met166. The tleap module of Amber14 (Case et al., 2015) was used to generate a periodic octahedral box containing the complex extended by 10 Å and filled with TIP3P water molecules and sodium ions to neutralize the overall charge of the system. Energy minimization and molecular dynamics (MD) simulation were performed with the parallel version of the pmemd module of Amber14. After an initial energy minimization with progressively reduced restraints on the position of all atoms (excluding solvent), 100 ps MD were run with temperature varying linearly from 0 to 300 K at constant volume, followed by a 400 ps MD at 300 K and a constant pressure of 1 bar, with the atomic coordinates saved every 2 ps. Three MD simulations of 20 ns were performed at a constant pressure of 1 bar, with atomic coordinates saved every 10 ps. The corresponding trajectories were concatenated in order to build a 60 ns MD simulation. Distance and angle variations in the course of the simulation were analyzed with the cpptraj module (Roe & Cheatham, 2013) from Amber14.”

Case, D.A., Berryman, J.T., Betz, R.M., Cerutti, D.S., Cheatham III, T.E., Darden, T.A., Duke, R.E., Giese, T.J., Gohlke, H., Goetz, A.W., Homeyer, N., Izadi, S., Janowski, P., Kaus, J., Kovalenko, A., Lee, T.S., LeGrand, S., Li, P., Luchko, T., Luo, R., Madej, B., Merz, K.M., Monard, G., Needham, P., Nguyen, H., Nguyen, H.T., Omelyan, I., Onufriev, A., Roe, D.R., Roitberg, A., Salomon-Ferrer, R., Simmerling, C.L., Smith, W., Swails, J., Walker, R.C., Wang, J., Wolf, R.M., Wu, X., York, D.M., and Kollman, P.A. (2015). AMBER 2015. (University of California, San Francisco)

Roe, D.R., and Cheatham, T.E., 3rd (2013). PTRAJ and CPPTRAJ: Software for Processing and Analysis of Molecular Dynamics Trajectory Data. J Chem Theory Comput 9:3084-95

- Page 6: “Molecular dynamics calculations performed on the structure suggest that this distorted arrangement is stable over the 60-ns simulation time, and that once trapped into such a conformation, no further fluctuation does occur (Supplementary Fig. 2a).”

- Page 9: “It is noteworthy that peptide G differs in conformation from peptides E and F to conserve the same type of interactions with the protein, and molecular dynamics calculations indicate that such binding mode is preserved during the 60 ns of simulation (Supplementary Fig. 2b).”
- Page 33: “Supplementary Figure 2: Molecular dynamics simulation. The variation of the root mean square deviation (rmsd) of the distance to the crystallographic structure is plotted along the 60 ns simulation. (a) rmsd computed independently for each protein chain of the *Mtb*-SecB^{TA} tetramer based on the C α atoms of residues found in secondary structure elements (see Fig. 1b). (b) rmsd computed independently for each ChAD peptide using all C α atoms.”
- Page 12: “Both Guinier and concentration-independent method (Rambo & Tainer, 2013) were applied to SAXS data and gave respective molecular weight of 84.5 ± 0.1 , 81.7 ± 0.4 kDa for *Mtb*-SecB^{TA} and 89.7 ± 0.1 , 83.6 ± 0.3 kDa for *Mtb*-SecB^{TA}/ChAD. The chaperone alone and its complex with ChAD form well folded molecular species, as can be observed from normalized Kratky plots (Fig. 4c, insert), and Guinier analysis gave Rg values of 3.17 ± 0.02 and 3.02 ± 0.02 nm and maximum dimensions of the particles (Dmax) of 9.5 ± 0.2 and 8.8 ± 0.2 nm, respectively.”
- Page 14: “The corresponding scattering curve is significantly distinct from those obtained for *Mtb*-SecB^{TA} alone or in complex with the peptide and reflects a well folded protein/protein complex (Fig. 4c) of molecular weight 114.4 ± 0.1 and 120.0 ± 0.3 kDa as obtained from Guinier and concentration-independent methods, respectively, and with Rg and Dmax values of respectively 3.61 ± 0.02 and 12.0 ± 0.2 nm.”

Another issue I have with the text in general, is the obsessive description of the details of the molecular interactions observed in the structures. From a mechanistic perspective, it is clear in some cases a detailed description of an interaction is required, however there are entire sections of the main manuscript that could be explained in a figure or are actually very well described in the figures of the paper (which I must say are very good). So I believe that the general reader of Nat Comm will appreciate if some of these parts are trimmed down.

This part of the manuscript and others have been substantially revised.

Major comments:

- The authors use the term, active antitoxin, what do they mean by that?

By active antitoxin, we meant an antitoxin capable of inactivating the toxin. We agree that this might not be appropriate for the antitoxin and have therefore replaced this term by “functional”.

- In some parts the authors used terms such as alpha-faces or alpha-interfaces that are not really corresponding to accepted structural features. Please revise that text throughout and correct this.

We removed the terms alpha-face and alpha-interface, beta-(inter)face as well, and replace them by dimer-dimer and dimer interface, respectively. This is actually the terminology used by Xu et al. and Deker et al. in their papers describing the structure of *Hi*-SecB and *Ec*-SecB, respectively.

- The lack or presence of electron density is described in several part of the text. In this regard the authors should provide an omit electron density map contoured around the region of the peptide. This is common practice in structural biology.

As requested by both reviewers 1 and 3, a new figure showing unbiased electron density map for all three peptides and in the vicinity of what would be a fourth binding site has been prepared (Supplementary Fig. 4).

- I find very interesting that the binding of the peptides to the chaperone is independent, specially since a transition from a dynamic to a more rigid state is involved. In this sense it would be interesting to do a reversed titration and measure the n value and K_D to validate that the interaction sites are indeed independent.

As suggested by reviewer 3, we performed reverse titration where the ChAD peptide (13.5 μM) in the cell was titrated by *Mtb*-SecB^{TA} (288 μM) (see Figure below). This experiment gave a stoichiometry of 2.2, thus consistent with a model preferentially involving two ChAD peptides per *Mtb*-SecB^{TA} tetramer and a K_D of 2.4 μM , in the same range than that determined by the different approaches mentioned in the manuscript (direct titration by ITC, MST, native MS). A sentence has been added in the revised version of the manuscript to mention that both direct and reverse ITC were performed.

Text modifications:

- Page 12: “Isothermal titration calorimetry (ITC), performed under both direct and reverse conditions, and microscale thermophoresis (MST) experiments showed that the chaperone interacts with the ChAD peptide with moderate affinity, with a K_D dissociation constant of $1.9 \pm 0.2 \mu\text{M}$ and $2.6 \pm 0.6 \mu\text{M}$, respectively (Fig. 4a,b).”

Titration of the ChaD peptide by *Mtb*-SecB^{TA}.

- Also regarding the thermodynamic measurements, I wonder why only two sites are detected when it seems apparent that at least 4 sites are easily detected by mass spectrometry, could the authors design a titration to measure the binding of 4 peptides?

This specific point is discussed in our response to point 4) raised by reviewer 1.

- The binding of the chaperone-antitoxin complex to DNA was also monitored by ITC, in this context it would be very very interesting if the authors could also measure the binding of the free antitoxin (or antitoxin DNA-binding domain) and the toxin-antitoxin to the same palindrome, in order to understand a bit better the interplay between all the players in transcription regulation.

Measuring the binding of the free antitoxin to DNA has also been proposed by reviewer 2 (point #10). Please see our detailed answer to reviewer 2 on pp. 8-9 of this document.

In addition, as suggested by reviewer 4, we confirmed by native MS experiments the formation of a chaperone/antitoxin/DNA heptameric 4:2:1 complex when mixing *Mtb*-SecB^{TA}/HigA1 and 38-bp palindromic sequence derived from the *higB* P2 promoter (see Supplementary Fig. 8a of the revised version).

- The dynamic state of the chaperone in the absence of the peptide as well as the folding state of HigA should be tested, given the relevance that the authors suggest this have. The dynamic state of the chaperone can not simply be assessed with these Kratky plots given that the overall structure of bound and unbound chaperone is very similar. Experiments of H-D exchange could help with this and if this is a very established fact of these type of protein it should be better explained in the text.

Following reviewer's suggestion, we have now performed hydrogen-deuterium exchange on *Mtb*-SecB^{TA} alone and in the presence of either ChAD peptide or *Mtb*-HigA1. These new findings are now included in the Results section, and a supplementary figure (Supplementary Fig. 7) has been added. Also sequence coverage and all kinetics of exchange (plots and .pml files) are now available for download as a zip file in the supplementary material (HDXMS-data.zip).

Text modifications:

- Pp. 12-13: **“To further investigate the dynamic behavior of the chaperone, we performed Hydrogen-Deuterium eXchange MS (HDX-MS) on *Mtb*-SecB^{TA} alone and in the presence of ChAD peptide. The kinetics of exchange for *Mtb*-SecB^{TA} alone revealed some highly dynamic and/or accessible regions (namely residues 13-18, 37-51, 59-67 and 161-181, in line with the absence of electron density for the C-terminal residues of *Mtb*-SecB^{TA}) compared to more protected regions (residues 68-75, 98-108, 124-152). Interestingly, a large decrease in deuteration upon binding of the ChAD peptide was observed for residues 21-27 and 153-159, which are precisely located along the peptide-binding pocket, and juxtaposing residues 52-57 (Supplementary Fig. 7a). These results also confirm that the chaperone is rather stable upon complex formation with ChAD.”**

- Page 14: **“Finally, we performed HDX-MS on the *Mtb*-SecB^{TA}/HigA1 complex. Finally, we performed HDX-MS on the *Mtb*-SecB^{TA}/HigA1 complex. Strikingly, binding of the chaperone to *Mtb*-HigA1 client resulted in the protection of the same stretches of**

residues as for *Mtb*-SecB^{TA}/HigA1 with the addition of residues 48-51 and 59-66, which are located in the continuity of the peptide-binding pocket (Supplementary Fig. 7b). This observation suggests a significantly larger binding interface between the chaperone and full-length *Mtb*-HigA1 when compared to the short ChAD peptide. We also did not find any specific destabilization of *Mtb*-SecB^{TA} upon complex formation with *Mtb*-HigA1, which is in agreement with the Kratky analysis of the SAXS data.”

- Page 25: “Automated Hydrogen-Deuterium eXchange coupled to Mass Spectrometry (HDX-MS). HDX-MS experiments were performed on a Synapt-G2 HDMS (Waters Scientific, Manchester, UK) coupled to a Twin HTS PAL dispensing and labelling robot (LEAP Technologies, Carborro, NC, USA) via a NanoAcquity system with HDX technology (Waters, Manchester, UK). Briefly, 5.2 μ L of protein at 15 μ M were diluted in 98.8 μ L of protonated (for peptide mapping) or deuterated buffer (20 mM MES pD 6.5, 200 mM NaCl) and incubated at 20 °C for 0, 0.5, 5, 10 and 30 min. 99 μ L were then transferred to vials containing 11 μ L of pre-cooled quenching solution (500 mM glycine at pH 2.3). After 30 s quench, 105 μ L were injected to a 100 μ L loop. Proteins were digested on-line with a 2.1 mm \times 30 mm EnzymateTM BEH Pepsin column (Waters Scientific, Manchester, UK). Peptides were desalted for 2 min on a C18 pre-column (Acquity UPLC BEH 1.7 μ m, VANGUARD) and separated on a C18 column (Acquity UPLC BEH 1.7 μ m, 1.0 \times 100 mm) by a linear gradient (2% to 40% acetonitrile in 13 min). Experiments were run in duplicates and protonated buffer was injected between each duplicate to wash the column and avoid cross-over contamination. Starting concentrations were 35 μ M for *Mtb*-SecB^{TA} alone and in the presence of 8 molar equivalent of ChAD peptide (280 μ M) and 18 μ M for *Mtb*-SecB^{TA}/HigA1. Peptide identification was performed with ProteinLynx Global SERVER (PLGS, Waters, Manchester, UK) based on the MSE data acquired on the non-deuterated samples. Peptides were filtered in DynamX 3.0 with the following parameters: peptides identified in at least 3 out of 5 acquisitions, 0.3 fragments per amino-acid, intensity threshold 1000.”

- Page 26: “Data availability. The data that support the findings of this study are available from the corresponding authors on request. The native MS and HDX-MS data are available as supplementary material.”

- Page 34: “Supplementary Figure 7: Hydrogen Deuterium eXchange MS. Differential hydrogen-deuterium uptake between (a) *Mtb*-SecB^{TA} and *Mtb*-SecB^{TA} incubated with 8 molar equivalents of ChAD or between (b) *Mtb*-SecB^{TA} and *Mtb*-SecB^{TA}/HigA1. The differential uptake is color-coded from blue (0%) to red (15-17%). Uncovered residues are shown in grey. The results at 30 min deuteration are plotted on the structure of *Mtb*-SecB^{TA}/ChAD. The ChAD peptides are represented as sticks.”

- In this sense, the authors state in the discussion that this is the first report of a SecB chaperone binding to a folded client. As it is, the statement is not properly supported by experimental data and is a bit misleading (for instance in the discussion the authors also refer to the toxin as “highly unstable”). While it is clear that HigA has a folded dimerisation domain that binds DNA, the folded state of the region that binds the chaperone is not described. Given that it has been shown that many antitoxins contain protein regions that are significantly disordered, the authors must show that this is not the case for this particular antitoxin or tone down this statement.

We agree with the reviewer: the term “highly unstable” used to describe the antitoxin *in vitro* is not appropriate since we only know that the antitoxin significantly aggregates and

is degraded by proteases (Bordes et al., 2011) in the absence of the chaperone and in the presence of the ChAD region. Here, we can simply discuss antitoxin functionality that is achieved in the presence of the chaperone, namely toxin inhibition and binding to DNA. We have thus toned down such a statement in the discussion replacing the term “highly unstable” by “aggregation-prone”, which so far reflects our *in vivo* and *in vitro* data.

Bordes, P., Cirinesi, A.M., Ummels, R., Sala, A., Sakr, S., Bitter, W., and Genevoux, P. (2011). SecB-like chaperone controls a toxin-antitoxin stress-responsive system in *Mycobacterium tuberculosis*. *Proc Natl Acad Sci U S A* 108:8438-43

Text modifications:

- Page 17: “To our knowledge, this is the first report of a SecB chaperone bound to a client protein that may have reached its functional form (Fig. 6). Whether such a conformation represents the fully folded antitoxin remains to be determined.”

- The proposed mechanism of the antitoxin release from the chaperone is one of the most exciting parts of the paper, given the implications this has in the regulation of the system. However the interaction of ChAD with the chaperone is on the weak side. In this regard it would be interesting if the authors could also measure the affinity of the chaperone for the full antitoxin or a longer version of the C-terminus of the antitoxin that is still soluble, to better understand how the ChAD peptide can titrate out the antitoxin.

The aggregative nature of HigA1 makes such experiments very challenging. Yet, we have performed ITC titration experiments as performed by Huang et al. *Nature* 537, 202-206 (2016) to measure the binding affinity of periplasmic maltose-binding protein (MBP) to *Ec*-SecB, where the slowly folding MBP^{V8G/Y283D} variant was unfolded in 8 M urea and diluted 20 times immediately before loading into the cell. In our case, urea-denatured *Mtb*-HigA1 (11 μM) was titrated by *Mtb*-SecB (131 μM), but this experiment did not give any indication of binding. We believe that in contrast with MBP^{V8G/Y283D}, the rapid aggregation of purified HigA1 upon dilution in urea might be responsible for such a behavior (Bordes et al., 2011). We next performed a second experiment where the chaperone (10 μM in 4 M urea) was titrated by *Mtb*-HigA1 (100 μM in 4 M urea). After buffer subtraction, a curve with a peculiar shape was obtained that could not be fitted with simple binding models (see Figure below). Several phases with sequential binding and 2-site binding could be identified but fitting such complex multi-site systems is quite hazardous and because no clear conclusion could be drawn from these data, we would prefer not include them in the manuscript. In addition, the high urea concentration used for the experiment could denature *Mtb*-SecB^{TA}.

Huang, C., Rossi, P., Saio, T., and Kalodimos, C.G. (2016). Structural basis for the antifolding activity of a molecular chaperone. *Nature* 537:202-6

ITC profile (integrated binding heats) of *Mtb*-HigA1 titration into *Mtb*-SecB^{TA} (4 M urea).

REVIEWER #4 (REMARKS TO THE AUTHOR) :

General Comment

The article presents a detailed structural and biophysical characterisation of key parts of a *Mycobacterium tuberculosis* SecB like toxin-antitoxin chaperone (TAC) system. The work concerns the SecB like chaperone (*Mtb*-SecBTA), that is known to bind a short region of the antitoxin (*Mtb*-HigA1), the "chaperone addition" sequence (ChAD). ChADh, destabilises the antitoxin in the absence of chaperone binding, thus rendering toxicity "chaperone mediated".

The authors here present detailed biophysical characterisation of *Mtb*-SecBTA both alone, and in complex with both ChAD, and also the full length antitoxin (*Mtb*-HigA1). New crystal structures of the *Mtb*-SecBTA tetramer and *Mtb*-SecBTA-3·ChAD are presented and compared in detail with previous structures of SecB homologs (*Ec*-SecB and *Hi*-SecB). The binding mode between ChAD and *Mtb*-SecBTA is compared and contrasted with published structures of SecB homologs bound to substrates. Detailed structural analysis and extensive mutagenesis is performed to identify the key binding residues and these mutants are tested *in vivo* for their toxin suppression ability in growth recovery assays.

Direct evidence of complex formation between *Mtb*-SecBTA and *Mtb*-HigA1 is presented using other biophysical methods, notably native MS, since this complex seems recalcitrant to crystallisation. *Mtb*-HigA1 is found to bind to *Mtb*-SecBTA as a dimer and this complex is found to bind a region of DNA *in vitro* corresponding to own repressor (repressor DNA binding is a known biological function). The authors use this as supporting evidence that the purified complex is in a functional state. Interestingly ChAD peptide is found to disrupt the interaction between HigA1 and *Mtb*-SecBTA.

The TAC system under study here and the results presented are fundamentally important in understanding the regulation of HigA1 in this human pathogen, and undoubtedly interesting for a specialist reader, however the reviewer is unsure of the more general significance and wider relevance. Results are presented clearly and with a high level of technical detail, however it is often unclear to a non-specialist what the purpose of the experiments are as you progress through the article other than presenting a detailed

biophysical/structural characterisation of the system. This kind of approach may be immediately tractable for a specialist audience, and suitable for a specialist journal, however, the reviewer suggests some reformatting and reorganising of the entire article, as well as a greater emphasis of putting the results into a wider biological context, would be necessary before publication in general interest a journal such as Nature Comm (see below for detailed comment).

Reformatting for a more general readership

Each section would strongly benefit from a more gradual introduction describing what the authors are investigating, and many sections would benefit from some concluding sentences or interpretation describing the relevance of data presented.

We agree with the reviewer and we have now introduced more biological significance and interpretation describing the relevance of the data (See also our answers below).

Instances of this occur throughout the article, for example..

>The Results section jumps straight into a detailed interpretation of the SecB structure. Given that the Introduction section that immediately precedes it culminates with a summary of the findings of the article, a few introductory sentences here would be extremely useful. Why are we suddenly concerned with the structure of SecB, what are the authors hoping to discover here?

As requested, we have now introduced the Results section, bringing in more details about what we were hoping to find and the main results that pushed us to solve such structure.

Text modifications:

- Page 5: “Three-dimensional structure of *Mtb*-SecB in complex with ChAD. In contrast with the classical export chaperone SecB, which binds and wraps unfolded presecretory protein substrates and maintains them in an unfolded conformation compatible with translocation through membrane (Huang et al., 2016), previous work revealed that the *Mtb*-SecB^{TA} chaperone of TAC is capable of binding to its aggregation-prone *Mtb*-HigA1 antitoxin substrate to facilitate, and thus not prevent, its folding in the cytosolic space. This strongly suggests a different binding mode of SecB-like chaperones with their antitoxin clients. To investigate such a novel SecB substrate-binding property, we have solved the structure of *Mtb*-SecB^{TA} together with the short C-terminal fragment of the *Mtb*-HigA1 antitoxin (named ChAD), which was found as the main site of interaction with the chaperone, both *in vivo* and *in vitro* (Bordes et al., 2016; Sala et al., 2017).”

Bordes, P., Sala, A.J., Ayala, S., Texier, P., Slama, N., Cirinesi, A.M., Guillet, V., Mourey, L., and Genevoux, P. (2016). Chaperone addiction of toxin-antitoxin systems. *Nat Commun* 7:13339

Huang, C., Rossi, P., Saio, T., and Kalodimos, C.G. (2016). Structural basis for the antifolding activity of a molecular chaperone. *Nature* 537:202-6

Sala, A.J., Bordes, P., Ayala, S., Slama, N., Tranier, S., Coddeville, M., Cirinesi, A.M., Castanie-Cornet, M.P., Mourey, L., and Genevoux, P. (2017). Directed evolution of SecB chaperones toward toxin-antitoxin systems. *Proc Natl Acad Sci U S A* 114:12584-9

>Section Mtb-SecB^{TA}/ChAD interaction. There is no introductory sentence here. What are the authors hoping to achieve by examining this interaction? A brief interpretation/summary at the end of the section would also be welcome.

As requested, we have now introduced the *Mtb*-SecB^{TA}/ChAD interaction part. A brief interpretation/summary at the end of this section would have focused on the fact that essential residues of *Mtb*-SecB^{TA} involved in the interaction with ChAD define two major binding spots. This is precisely developed in the following section and a comparison with respect to the client binding site in SecB has been made in the Discussion.

Text modifications:

- Page 9: “*Mtb*-SecB^{TA}/ChAD interaction. In order to investigate how the chaperone specifically recognizes the C-terminal chaperone-addiction region of the *Mtb*-HigA1 antitoxin, key residues involved in the interaction between the ChAD peptide (*Mtb*-HigA1₁₀₄EVPTWHRLSSYRG₁₁₆) and the *Mtb*-SecB^{TA} chaperone substrate-binding surface were analyzed in detail.”

- Page 10: “Mutagenesis of *Mtb*-SecB^{TA} residues involved in ChAD binding. In order to investigate whether the newly discovered interaction between *Mtb*-SecB^{TA} and the antitoxin was relevant *in vivo*, all residues of *Mtb*-SecB^{TA} involved in polar interactions with the ChAD peptide, either through side-chain or main-chain atoms, and those involved in non-bonding interactions through burying more than 50% of their solvent accessible surface (Fig. 1b and Supplementary Table 2) were systematically mutated to alanine residues.”

>The section “Comparison of Mtb-SecB structures” contains a very detailed comparison of the respective crystal structures, but very little interpretation as to the biological or functional significance of the differences described. This would be extremely helpful, especially for a general readership.

As requested, we have added some biological and functional significance to the structural data, without being too speculative. Furthermore, as requested by all reviewers, this part and others have been substantially shortened or revised.

Text modifications:

- Page 8: “Whether these structural differences are important for TA control, export function or protein stability is unclear. The fact that all the TAC chaperones tested so far could efficiently replace *Ec*-SecB export function, regardless of the sequence similarities (Bordes et al., 2016; Sala et al., 2013), and that single point mutations in substrate-binding area were sufficient to redirect *Ec*-SecB toward *Mtb*-HigBA1 without affecting its function in protein export (Sala et al., 2017), suggest that such differences may not be critical for TA control or export.”

- Page 8: “For instance, there is a strong electronegative cluster right in the middle of the SecB β -sheet, which arises from side-chain clustering of residues D20/D27, E24/E31, and E77/E86 (*Ec*-SecB/*Hi*-SecB numbering), which is important for interaction with SecA C-terminal region. In *Mtb*-SecB^{TA}, these residues are replaced by R32, A36 and D102, respectively. The fact that *Mtb*-SecB^{TA} is less efficient than *Ec*-SecB when tested for SecA-

dependent function in *E. coli* suggests that such modified electrostatic interface might be responsible for the reduced complementation *in vivo*.”

Bordes, P., Sala, A.J., Ayala, S., Texier, P., Slama, N., Cirinesi, A.M., Guillet, V., Mourey, L., and Genevoux, P. (2016). Chaperone addiction of toxin-antitoxin systems. *Nat Commun* 7:13339

Sala, A., Calderon, V., Bordes, P., and Genevoux, P. (2013). TAC from *Mycobacterium tuberculosis*: a paradigm for stress-responsive toxin-antitoxin systems controlled by SecB-like chaperones. *Cell Stress Chaperones* 18:129-35

Sala, A.J., Bordes, P., Ayala, S., Slama, N., Tranier, S., Coddeville, M., Cirinesi, A.M., Castanie-Cornet, M.P., Mourey, L., and Genevoux, P. (2017). Directed evolution of SecB chaperones toward toxin-antitoxin systems. *Proc Natl Acad Sci U S A* 114:12584-9

> Section “Specific affinity of ChAD for Mtb-SecB^{TA} chaperone”. The author assumes that given the high-resolution crystal structures presented in previous sections, the purpose of this section is to characterise the interactions “in-solution” and build upon the static “snapshot” data already presented? Perhaps an introductory sentence to guide the reader would help here?

The introductory sentence has been modified as suggested by the reviewer.

Text modifications:

- Page 12: “Specific affinity of ChAD for *Mtb*-SecB^{TA} chaperone. Several approaches were used to characterize complex formation between *Mtb*-SecB^{TA} and the ChAD peptide in solution and to further confirm structure-function relationships deduced from the X-ray structure and mutagenesis experiments.”

>Final paragraph - “We next used Native MS to investigate whether a stably formed... complex is could be disrupted by increasing concentrations of ChAD peptide.” What is the biological relevance of this experiment, what are the authors trying to demonstrate by it and on what basis?

We have addressed this specific point by introducing the rationale behind these experiments.

Text modifications:

- Page 15: “It has been proposed that disrupting the TA interfaces by small molecules or peptides to trigger endogenous toxin activation *in vivo* could represent a novel antibacterial tool to induce bacterial death from the inside (Chan et al., 2015). The tripartite TAC system significantly differs from classical two-component TA systems due to the fact that the TAC antitoxin is significantly destabilized by the acquired C-terminal ChAD segment, which efficiently triggers antitoxin aggregation and/or degradation in the absence of chaperone, thus leading to toxin activation. Therefore, we hypothesized that disrupting an earlier step in the toxin inactivation pathway, namely the interaction between the chaperone and its antitoxin client, might lead to growth inhibition. To begin to approach this specific point, we first used native MS experiments to investigate whether

a stably formed *Mtb*-SecB^{TA}/HigA1 complex could be disrupted *in vitro* by increasing concentrations of ChAD peptide.”

Chan, W.T., Balsa, D., and Espinosa, M. (2015). One cannot rule them all: Are bacterial toxins-antitoxins druggable? FEMS Microbiol Rev 39:522-40

>Discussion, the comparison of the binding mode between *Mtb*-SecB and other SecB homologs is made. How significant/surprising are the differences observed the given the different biological functions and specificities of the chaperones

We believe that this work reveals for the first time that the binding mode of SecB homologues to a TAC antitoxin significantly differs from what was known before. Indeed, we showed both *in vivo* and *in vitro* that a SecB chaperone can facilitate the folding of a cytosolic substrate by binding to a specific region that strongly destabilizes the substrate. This is in sharp contrast with the canonical role of SecB as antifolding chaperone, where it binds and wraps large peptide segments of its client to allow their subsequent translocation through the Sec pore. Because SecB from TAC systems can perform protein export and control their respective TA systems (Bordes et al, 2016; Sala et al., 2017), we propose that both functions could be coupled, which might represent a yet unknown link between toxin activation and protein export stress. We have now further highlighted these specificities in the Discussion section.

Text modifications:

- Page 17: “To our knowledge this is the first report of a SecB chaperone bound to a client protein that may have reached its functional form (Fig. 6). Whether such a conformation represents the fully folded antitoxin remains to be determined. Although SecB chaperones mainly assist protein export by preventing the folding of presecretory protein clients, our data show that some SecB chaperones can also assist the folding of cytosolic substrates such as the *Mtb*-HigA1 antitoxin. This property is in agreement with the observation that overexpressed *Ec*-SecB can partially rescue the bacterial growth defect of an *E. coli* strain lacking both major cytosolic chaperones DnaK and Trigger Factor, and that *Ec*-SecB can bind cytosolic proteins co- and/or post-translationally *in vitro* (Ullers et al., 2004).

Ullers, R.S., Luirink, J., Harms, N., Schwager, F., Georgopoulos, C., and Genevaux, P. (2004). SecB is a bona fide generalized chaperone in Escherichia coli. Proc Natl Acad Sci U S A 101:7583-8.

Detailed comments

Introduction

Page 1 - “(in about 7% of the cases)” change to “(in about 7% of cases)”

Page 1 - “It is believed that under certain stress the less stable...” suggest change to “It is believed that under certain stress conditions the less stable...”

Page 1 - “ ...thus inhibiting growth until normal growth condition resume.”

Change to “ ...thus inhibiting growth until normal growth conditions resume.”

Or reword

All changes have been made accordingly.

Page 2 - sentence on "drug persisters" - this is an important point to demonstrate the wider relevance of the work presented. The article may benefit from describing the role of TAC in drug resistance in more detail here.

As requested, we have given more details about the relevant stress conditions in which TAC is induced, including antibiotic-induced persistence, without being too speculative.

Text modifications:

- Page 4: "Noticeably, *Mtb*-TAC genes were shown to be significantly induced in response to several stresses relevant for *M. tuberculosis* pathogenesis. This includes DNA damage (Smollett et al., 2009), heat shock (Stewart et al., 2002), nutrient starvation (Betts et al., 2002), hypoxia (Ramage et al., 2009), host phagocytes (Tailleux et al., 2008) and antibiotic-induced persistence (Keren et al., 2011), suggesting that the TAC system could contribute to the stress adaptive response and/or to the virulence of the bacillus. Yet, such a role for TAC remains to be demonstrated."

Betts, J.C., Lukey, P.T., Robb, L.C., McAdam, R.A., and Duncan, K. (2002). Evaluation of a nutrient starvation model of Mycobacterium tuberculosis persistence by gene and protein expression profiling. Mol Microbiol 43:717-31

Keren, I., Minami, S., Rubin, E., and Lewis, K. (2011). Characterization and transcriptome analysis of Mycobacterium tuberculosis persisters. MBio 2:e00100-11

Ramage, H.R., Connolly, L.E., and Cox, J.S. (2009). Comprehensive functional analysis of Mycobacterium tuberculosis toxin-antitoxin systems: implications for pathogenesis, stress responses, and evolution. PLoS Genet 5:e1000767

Smollett, K.L., Fivian-Hughes, A.S., Smith, J.E., Chang, A., Rao, T., and Davis, E.O. (2009). Experimental determination of translational start sites resolves uncertainties in genomic open reading frame predictions - application to Mycobacterium tuberculosis. Microbiology 155:186-97

Stewart, G.R., Wernisch, L., Stabler, R., Mangan, J.A., Hinds, J., Laing, K.G., Young, D.B., and Butcher, P.D. (2002). Dissection of the heat-shock response in Mycobacterium tuberculosis using mutants and microarrays. Microbiology 148:3129-38

Tailleux, L., Waddell, S.J., Pelizzola, M., Mortellaro, A., Withers, M., Tanne, A., Castagnoli, P.R., Gicquel, B., Stoker, N.G., Butcher, P.D., Foti, M., and Neyrolles, O. (2008). Probing host pathogen cross-talk by transcriptional profiling of both Mycobacterium tuberculosis and infected human dendritic cells and macrophages. PLoS One 3:e1403

Page 2 - "...RelE toxin superfamily of ribonuclease." Change for "...RelE toxin superfamily of ribonucleases."

Done.

Results

Section on Mtb-SecBTA/ChAD interaction. Only 3 ChAD are found bound in this highly symmetrical structure rather than the expected 4, could the authors speculate as to why this occurs?

Similar question for Calcium, that is found in 3 but not 4 subunits?

This is now discussed in several instances, see for example our responses to reviewer 1.

Section "Specific affinity of ChAD for Mtb-SecBTA chaperone". The reviewer agrees with the author that the binding of higher numbers of ChAD is likely non-specific, especially due to the high concentrations of ChAD used in binding experiments here (0-195uM from supp info). It should be noted somewhere in the article/supp, that concentrations of substrate+enzyme > ~50uM likely lead to non-specific adduction in the electrospray process. Ref - "Protein Complexes in the Gas Phase: Technology for Structural Genomics and Proteomics" Chem. Rev., 2007, 107 (8), pp 3544-3567 and others.

The text has been modified accordingly, as proposed by reviewer 4.

Text modifications:

- Page 13: "This low-abundance interaction may be related to some non-specific binding also observed at high molar ratios of ChAD peptide. The non-specific adduction is a common feature of native MS when working with substrate/ligand concentrations greater than 50 μ M (Benesch et al., 2007)."

Benesch, J.L., Ruotolo, B.T., Simmons, D.A., and Robinson, C.V. (2007). Protein complexes in the gas phase: technology for structural genomics and proteomics. Chem Rev 107:3544-67

Correction for this has been proposed when calculating KD values "A method for removing effects of nonspecific binding on the distribution of binding stoichiometries: application to mass spectroscopy data." Biophys J. 2010 Sep 8;99(5):1645-9. The authors wondered if this correction could be applied to test for, and quantify, the non-specific adduction in their data? And even perhaps calculate more accurate kD values?

We have tried to apply this method to estimate nonspecific binding and derive more accurate values for the specific binding constants. We managed to estimate a nonspecific binding K_D of 480 μ M based on the ratio of intensities of the fifth, sixth and seventh binding events. Although in the good logarithmic range (1-60 μ M), K_D values estimated this way for the specific binding events were somewhat surprising: K_{D1} =9.4 μ M, K_{D2} =19.4 μ M, K_{D3} =64.5 μ M, K_{D4} =1.14 μ M. We think that the fitting of the data performed by Unidec on the entire titration is, at least in our case, more relevant. The fact that the estimation of the nonspecific binding K_D is two orders of magnitude higher suggests that it does not affect so much our calculated values.

The results state that KD's of 9-23 uM could be calculated. Could the authors please confirm if these are microstate kDs and if so add the used in Unidec to calculate them including the model used and justification, to the supp info.

These are indeed microstate dissociation constants. They were calculated in Unidec with the following parameters: How to extract: Local Max, Number of proteins: 1 (we used the concentration of tetramers), Number of Ligands: 4; Protein and Ligand Models: All K_D 's Free; Binding Sites Per Proteins: 4. These parameters have been included in the Methods section and the values obtained with the new data processing parameters (see below) are now detailed in the Results section: K_{D1} =6.1 μ M, K_{D2} =8.8 μ M, K_{D3} =7.2 μ M and K_{D4} =16.5 μ M.

Text modifications:

- Page 13: “Plotting the relative abundance obtained for each species at increasing molar ratios and fitting the corresponding curves (Supplementary Fig. 8b) confirmed this observation, with dissociation constant values $K_{D1}=6.1 \mu\text{M}$, $K_{D2}=8.8 \mu\text{M}$, $K_{D3}=7.2 \mu\text{M}$ and $K_{D4}=16.5 \mu\text{M}$.”
- Page 25: “ K_D values were obtained by fitting titration curves in the data collector utility from Unidec with the following parameters: “How to extract”: Local Max, “Number of proteins”: 1 (tetrameric concentration), “Number of Ligands”: 4, “Protein and Ligand Models”: All K_D ’s Free, “Binding Sites Per Proteins”: 4.”

For the native MS data, the deconvoluted spectra show binding to the tetramer and hexameric complex in Figs 4 S4 and S5. The spectra in these figures do not cover the mass range required to also see the monomer/lower order oligomers that may also be present. Could the authors confirm if any of these lower order oligomers were observed and if so how binding of ChAD to them was accounted for in the data analysis?

Concerning the titration of *Mtb*-SecB^{TA} with ChAD, we observed that dissociation into monomeric subunits in the low m/z range is very limited and accounts for less than 10% of the signal, even using a sample cone of 150 V (see comments below). With respect to the titration of *Mtb*-SecB^{TA}/HigA1 with ChAD, the proportion of dissociated SecB was equivalent but the proportion of HigA1 was higher (around 30-50% of the total signal), meaning that the *Mtb*-SecB^{TA}/HigA1 hexamer is less stable than the *Mtb*-SecB^{TA} tetramer in the gas phase. However, we checked by SEC-MALS that the hexamer is extremely stable in both purification and native MS buffers. In addition, *Mtb*-HigA1 was found to be highly insoluble. Thus, we believe that we really focused here on the increase of dissociation triggered by ChAD titration/binding.

The reviewer strongly suggests that the raw data be made available for examination by the reader in any final publication.

The raw data used in native MS and HDX analyses and for producing Fig. 4, Fig. 5, Supplementary Fig. 4 (Supplementary Fig. 8 in the revised version), and Supplementary Fig. 5 (Supplementary Fig. 9 in the revised version) are now available for download as zip files in the supplementary material (NativeMS-data.zip and HDXMS.zip).

Brief question, did the native MS results indicate the presence of calcium as suggested by the crystal structure? (the reviewer acknowledges that this may not have been resolvable)

Calcium ions observed in the X-ray structure were from the crystallization medium and we indeed did not see them in native MS using purified protein from the SEC elution. We performed additional experiments whereby *Mtb*-SecB^{TA} or *Mtb*-SecB^{TA}/ChAD were desalted in calcium acetate up to a concentration of 360 mM, i.e. higher than that used for crystallization, but could not see the presence of any Ca²⁺ adducts. Indeed, when looking at smoothed but non-deconvoluted data, we can clearly see classical +22 Da sodium adducts on the *Mtb*-SecB^{TA} charge states. However, no clear evidence for +40 Da calcium adducts were visible, even when considering that these adducts could overlap with half of the Na adducts (even numbers).

Section "Full length MtB-SecTA/HigA1 chaperone/antitoxin complex". Authors use SEC-MALS to confirm interaction of MtB-SecTA/HigA1 hexamer with DNA, which likely forms a 4:2:1 complex and also confirm by ITC which indicates an association of 1.1. Could the authors confirm the error in the later (1.1)? Was it possible to verify this result using native MS which would provide much more certainty to the complex stoichiometry and presumably indicate if indeed there is "binding of slight more than one palindromic DNA sequence per MtB-HigA1 dimer."

We confirmed the stoichiometry of 1.1. As suggested, we used native MS to analyze *Mtb-SecB^{TA}/HigA1* in complex with DNA (i.e. the 38-bp palindromic sequence derived from the *higB* P2 promoter) and observed that they indeed form a 4:2:1 complex (MW=138.25 kDa). This result is now part of Supplementary Fig. 9 (revised version numbering) (panel a) and the Methods and Results sections have been updated accordingly.

Text modifications:

- Page 14: "We also used native MS to analyze *Mtb-SecB^{TA}/HigA1* in complex with its specific DNA target and observed that they indeed form a 4:2:1 complex of molecular weight 138.25 kDa (Supplementary Fig. 9a)."

- Pp. 34-35: "Supplementary Figure 9: DNA-binding properties and disruption by ChAD of *Mtb-SecB^{TA}/HigA1*. (a) Deconvoluted native mass spectrum of *Mtb-SecB^{TA}/HigA1* in the presence of dsDNA derived from the *higB* P2 promoter . (b-f) Deconvoluted native mass spectra of *Mtb-SecB^{TA}/HigA1* showing how the *Mtb-SecB^{TA}/HigA1* hexamer is progressively destabilized/dissociated by increasing concentrations of the ChAD peptide. Circles and squares indicate ChAD peptides bound to the *Mtb-SecB^{TA}* tetramer and *Mtb-SecB^{TA}/HigA1* hexamer, respectively (same color code of the symbols as in Supplementary Fig. 8)."

Final paragraph. The authors interpret the dissociation of the MtB-SecTA/HigA1 hexamer upon addition of ChAD, to binding to the hexamer, followed by dissociation. Given that much binding in the native MS experiments may be non-specific, could the authors please justify their interpretation here? Another explanation, that may be equally likely given the data presented, is that since the MtB-SecTA/HigA1 exists in equilibrium with free- MtB-SecTA, addition of ChAD may also bind directly to MtB-SecTA. If this MtB-SecTA.ChAD cannot bind HigA1 this would likely shift the MtB-SecTA/HigA1 vs free-MtB-SecTA equilibrium towards the tetramer thus favouring hexamer dissociation. Could the authors comment in the text on whether this mechanism may also occur?

This particular point is discussed at the top of the previous page.

Native MS Methods

The cone voltage of 150V seems very high for native measurements, and especially for binding measurements could the authors please comment?

As usual in native MS, several parameters had to be optimized, including the cone voltage, in order to find a good compromise between transmission (intensity of the signal) and resolution (quality of the signal, narrowness of the peaks). In the case of *Mtb-SecB^{TA}*, we started with a value of 150 V and manually lowered it to 25 V with 10 V increments. We finally chose a cone voltage of 150 V as the quality of the peaks corresponding to the

tetramer was best at this value, with only limited dissociation into monomers compared to lower values.

The cap voltage is said to be 1.6kV is this in addition to the potential applied directly to the nanomate chip?

The 1.6 kV was indeed applied directly to the nanomate and in this set-up, the capillary voltage of the instrument was not applicable (and set to 0 V anyway). The authors referred to this voltage as the “capillary voltage” but agree that it could be misleading. The term “capillary voltage” was thus replaced by “the voltage applied to the chip”.

Subtract curve in UniDec is known to potentially greatly distort intensities of CS distributions in distinct m/z ranges, present in the same spectrum (such as the tetramer/ hexamer here). The value for this parameter should remain quite constant throughout the same data set for fair interpretation and comparison. Have the authors examined the effect of this parameter on relative intensities and perhaps performed manual validation on at least one data point to check appropriateness? Using a FWHM peak width of 1Th will likely hugely overfit the data (peaks on Q-ToF more likely 10-100Th wide) resulting in intensity abnormalities, the reviewer suggests using a more realistic peak width for data processing.

The reviewer is right in saying that the subtract curve and FWHM parameters are key to the deconvolution process in Unidec, which is why a great care was taken. In the submitted version, these parameters were optimized for each raw spectrum. Background subtraction was actually only necessary for the highest points of the titration because of higher baseline levels. However, we understand the concerns raised by reviewer 4 and the data were re-analyzed using exactly the same data processing parameters for all the spectra used in this study (Subtract curve=5 and FWHM=10). The changes were included in the new version of the manuscript: the new parameters were added to Methods section, the figures were amended with the new deconvoluted spectra and the new K_D values (that are only slightly modified: 6.1/8.8/7.2/16.5 μM) were added in the Results section.

REVIEWERS' COMMENTS:

Reviewer #2 (Remarks to the Author):

From my point of view the new version of the manuscript reads smoother and it is clearer.

Authors addressed all my points/questions in a detailed/rigorous manner.

I have no further questions

Damián Lobato-Márquez

Reviewer #3 (Remarks to the Author):

The authors addressed most of the concerns and questions raised by the reviewers. The revised manuscript considerably improved from the inclusion of additional data, enhancing the quality of figures and editing. Unfortunately, the manuscript still suffers from the misused of certain terms that should be resolved before it is published in Nature Communications (see my comments below for these minor remarks).

Line 50, page 3: Please provide a reference for the 7% figure that is provided referring to the frequency of TAC systems.

Line 59, page 3: Please correct, TA systems are widely distributed in mobile genetic elements in general (not only plasmids, they are presents also in phages, transposons, insertion sequences...) Throughout the manuscript the term "stability" is confused with aggregation propensity. These are different things, the antitoxin has a high aggregation propensity as observed by the authors, however the stability of the protein which refers to its free energy was never measured, please address and correct this issue. As an example in line 83, page 4, it is stated that the ChAD peptide destabilises the antitoxin, which is not supported by the experimental data. The experiments show that the ChAD peptide makes the antitoxin more aggregation prone.

Line 89, page 4: It is pure speculation to say that the observation that SecB chaperones retain their ability to export, suggests that the TAC activation cycle could be linked to protein export. Please remove this statement or provide adequate experimental evidence.

Line 122, page 5: "a long 30-residues long C-terminal..." should be "a 30-residues long C-terminal..."

Please change the term beta-dimer to just dimer (a beta-dimer has no structural meaning)

Line 156, page 6: Ca²⁺ entities should be changed to "Ca²⁺ ions"

Line 166, page 7: "ca" should be changed to "C-alpha"

How reliable are the K_d measured with mass spectrometry? What is the error of the measurement (this is important given that in some cases the curves contained only a few points). In general none of the K_d are given with the proper measurement error and binding statistics are in general lacking, this should be provided in the final version of the paper.

Line 449, page 15: "run" should be changed to "runs"

Reviewer #4 (Remarks to the Author):

The authors have adequately and conscientiously addressed all comments and have substantially reformatted the paper. I am now satisfied that it is more suitable for a general audience and would be happy to recommend its acceptance for publication.

REVIEWER #3 (REMARKS TO THE AUTHOR)

The authors addressed most of the concerns and questions raised by the reviewers. The revised manuscript considerably improved from the inclusion of additional data, enhancing the quality of figures and editing. Unfortunately, the manuscript still suffers from the misused of certain terms that should be resolved before it is published in Nature Communications (see my comments below for these minor remarks).

Line 50, page 3: Please provide a reference for the 7% figure that is provided referring to the frequency of TAC systems.

The 7% figure is associated with reference #10 added at the end of the sentence.

Line 59, page 3: Please correct, TA systems are widely distributed in mobile genetic elements in general (not only plasmids, they are presents also in phages, transposons, insertion sequences...)

We have adapted this sentence as requested on lines 57-58: "TA systems are also widely distributed on mobile genetic elements, including plasmids, where they contribute to their stability."

Throughout the manuscript the term "stability" is confused with aggregation propensity. These are different things, the antitoxin has a high aggregation propensity as observed by the authors, however the stability of the protein which refers to its free energy was never measured, please address and correct this issue. As an example in line 83, page 4, it is stated that the ChAD peptide destabilises the antitoxin, which is not supported by the experimental data. The experiments show that the ChAD peptide makes the antitoxin more aggregation prone.

We agree with the reviewer and have now corrected this throughout the manuscript: lines 18, 81, 397 and 1044.

Line 89, page 4: It is pure speculation to say that the observation that SecB chaperones retain their ability to export, suggests that the TAC activation cycle could be linked to protein export. Please remove this statement or provide adequate experimental evidence.

As requested, we have deleted this sentence.

Line 122, page 5: "a long 30-residues long C-terminal..." should be "a 30-residues long C-terminal..." Please change the term beta-dimer to just dimer (a beta-dimer has no structural meaning)

Done.

Line 156, page 6: Ca²⁺ entities should changed to "Ca²⁺ ions"

Done.

Line 166, page 7: "ca" should be changed to "C-alpha"

Done.

How reliable are the K_d measured with mass spectrometry? What is the error of the measurement (this is important given that in some cases the curves contained only a few points). In general non of the K_d are given with the proper measurement error and binding statistics are in general lacking, this should be provided in the final version of the paper.

The microstate K_d values were estimated by the Unidec software dedicated to the analysis of native MS data. This software does not provide standard deviation values unless the experiments are run in multiplicate. This is why in our first draft submission we wrote:

"Plotting the relative abundance obtained for each species at increasing molar ratios and fitting the corresponding curves (Supplementary Fig. 4b) confirmed this observation, with K_d values in the range 9-23 μ M."

At this stage, we wanted to emphasize that the K_d s obtained by ITC and native MS were in the same micromolar range. It is only to answer to a question raised by reviewer #4 that we gave the values for each microstate K_d . We would agree to go back to the earlier version stating only the range of these K_d s, and leave this decision to the Editor.

Line 449, page 15: "run" should be changed to "runs"

"run" seems appropriate as the subject of the sentence is "the ChAD peptide binding sites".